# Two-Layer Convolutional Autoencoders Trained on Normal Data Provably Detect Unseen Anomalies

**Yanbo Chen, Weiwei Liu**[*]
School of Computer Science, Wuhan University, Wuhan, China
{yanbo.acad,liuweiwei863}@gmail.com

## Abstract

Anomaly detection refers to the techniques that identify (probably unseen) rare or suspicious data that deviate significantly from the pre-defined normal data (Chalapathy & Chawla, 2019; Ruff et al., 2021). Empirical studies have observed that generative models trained on normal data tend to produce larger reconstruction errors when reconstructing anomalies. Based on this observation, researchers have developed various anomaly detection methods, referred to as reconstruction-based anomaly detection (RBAD) (Lv et al., 2024; Li et al., 2024) in the literature.

Despite the empirical success of RBAD, the theoretical understanding of RBAD is still limited. This paper provides a theoretical analysis of RBAD. We analyze the training dynamics of a 2-layer convolutional autoencoder and introduce the *cone set* of the features. We prove that the cone sets of the normal features would *absorb* the (convolutional) kernels of the autoencoder during training and use these absorbed kernels to reconstruct the inputs. The absorbed kernels are more aligned to the normal features, which explains the cause of the reconstruction error gap between the normal data and the anomalies. Synthesized experiments are provided to validate our theoretical findings. We also visualize the training dynamics of the autoencoder on real-world data, demonstrating our proposed cone set intuition.

## 1 Introduction

Anomalies are rare or suspicious observations that deviate significantly from the pre-defined normal data (Chalapathy & Chawla, 2019; Ruff et al., 2021). Examples of anomalies include credit card fraud (Bhattacharyya et al., 2011), defects in manufacturing products (Bergmann et al., 2021), and outbreaks of disease (Díaz-Cao et al., 2023). Anomaly detection (AD) is a challenging task since negative samples (i.e., anomalies) are hard to obtain at the training stage. Nevertheless, researchers have developed various AD methods that require only normal training data. For example, empirical studies (Salehi et al., 2021; Mousakhan et al., 2023) have observed an intriguing phenomenon that generative models trained on normal data tend to produce larger reconstruction errors when reconstructing anomalies than normal data. Accordingly, anomalies can be detected by setting up thresholds for reconstruction errors. Such AD methods are referred to as reconstruction-based anomaly detection (RBAD) (Livernoche et al., 2023; Lv et al., 2024; Li et al., 2024) in the literature.

Despite the empirical success of RBAD, the theoretical understanding of RBAD is still limited. A detailed discussion on the related works is postponed to Appendix A due to limited space. The main goal of our paper is to establish a theoretical framework for analyzing RBAD and find a plausible explanation for RBAD. More specifically, we raise the following three research questions.

- ($Q_1$) *How does the model learn the features from normal training data in RBAD?*
- ($Q_2$) *What is the role of the learned features in the reconstruction phase?*
- ($Q_3$) *Why are the anomalies harder to reconstruct compared to normal data?*

This paper comprehensively discusses $Q_1$ - $Q_3$ and provide a brief answer in Section 5.

---

[*]Correspondence to: Weiwei Liu {liuweiwei863@gmail.com}.

**Setup.** We analyze the training and reconstructing processes of RBADs using tools from *feature learning* (Allen-Zhu & Li, 2021; 2023); we refer the readers to Appendix A for more related works. Following Allen-Zhu & Li (2023), we characterize the inputs images as an array of $P$ *patches*. As illustrated in Figure 1, patches are small parts of an image that might contain various features and non-feature noise. In this paper, we adopt the signal-noise data model that characterizes the patches as the sum of a feature and a random noise. For normal data, the patches contain different features according to some given distribution (cf. Section 2.1). We characterize the anomalies as normal data with some of the normal patches replaced by abnormal patches.

We consider a *two-layer convolutional autoencoder* that includes a convolutional hidden layer with max pooling operation. The network is parameterized by the convolutional kernels $w$ and is trained by minimizing the empirical reconstruction error using gradient descent with random initialization. We analyze the training dynamics of the kernels by setting up probability bounds for the growth rates and the magnitude of the kernels.

**Theoretical Analysis.** We quantitatively analyze the training and reconstruction phases of RBADs in Section 3. Given feature and iteration step, Definition 3.1 introduces the *cone set* of the feature at that iteration step. By definition, the "cones" would get sharper as iteration steps increase (cf. Figure 3), which ensures that the kernels in the cone sets would get more aligned to the corresponding feature. We prove that the cone sets of the features would absorb the kernels during training. As a result, the absorbed kernels would eventually approximates the features' directions, and by this means, the model extract and learn the features from the training data. The above analysis provides a plausible answer to question $Q_1$.

We answer the questions $Q_2$ and $Q_3$ by further analyzing the reconstruction phase of RBAD. Our paper characterizes the anomalies as normal data with some of the normal patches replaced by the abnormal patches. In Section 3.2, we study how the learned features process the signal from the input image. Theorem 3 proves that the processed signal of the abnormal patches would be significantly weaker than those of the normal patches, explaining why the anomalies are hard to reconstruct.

**Empirical Validation.** We conduct experiments on synthesized and real-world datasets to validate the cone set intuition. The setting of the synthesized experiment is identical to our theoretical setting. As reported in Figure 4, there is a clear gap between the reconstruction errors of the normal data and the anomalies, which is in consistent with both our theoretical claims and the existing empirical results on RBAD. We also train networks on real-world anomaly detection datasets and visualize the growth of the kernels in these networks. The results exhibit clear feature learning processes, which further validate our proposed cone set intuition.

**Contribution Statement.** The main contributions of our paper are summarized as follows.

1. We establish a theoretical framework for analyzing RBAD. The main results and intuitions are summarized in the theoretical roadmap (Figure 2). Our proposed framework introduces the cone set intuition (Remark 3.2 and Figure 3). We characterize how neural networks extract features ($Q_1$) and reconstruct normal data and anomalies ($Q_2$). In particular, we prove that a 2-layer convolutional autoencoder trained on normal data cannot accurately reconstruct the anomalies with high probability ($Q_3$).

2. We conduct experiments on synthesized and real-world datasets to validate our theoretical findings. With an identical setting to our theory, the results of the synthesized experiment are consistent with both our theoretical findings and the existing empirical results. We also visualize the feature learning process on real-world datasets, validating our proposed cone set intuition. **Remark:** It is worth noting that the main goal of this paper is to *complement*, not *compete* with, existing RBAD methods. The experiments are mainly conducted to support and validate our theoretical findings.

The remainder of this paper is organized as follows. Section 2 introduces the problem setup and the terms and notations. A detailed discussion on the related works is postponed to Appendix A. We present the main results and intuitions in Section 3, and postpone the detailed proofs to Appendix E. Experimental results and discussions can be found in Section 4. We conclude this paper in Section 5.

## 2 PROBLEM SETUP

Throughout this paper, we denote the vectors by bold letters (e.g., $\boldsymbol{x}$, $\boldsymbol{v}$, and $\boldsymbol{w}$). The $i$-th entry of $\boldsymbol{x}$ is noted by $\boldsymbol{x}^{(i)}$. For any $N \in \mathbb{Z}^+$, we write $[N] := \{1, 2, \cdots, N\}$ and $[\boldsymbol{x}_N] = \{\boldsymbol{x} : \exists i \in [N] \ s.t. \ \boldsymbol{x} = \boldsymbol{x}_i\}$ for simplicity. The 2-norm of $\boldsymbol{x}$ is denoted by $\|\boldsymbol{x}\|$. A summarized list of terms and notations is included in Appendix C.

**$P$-Patch Data.** Following Allen-Zhu & Li (2023), we consider the following $P$-patch data ($P \in \mathbb{Z}^+$) model. As demonstrated in Figure 1, a patch $\boldsymbol{x}$ refers to a small part of an image $\boldsymbol{z}$ that might contain various features (e.g., edges and shapes). Previous works of feature learning (Allen-Zhu & Li, 2023; Kou et al., 2023) characterize the patches as vectors. In this paper, we let $\mathcal{H}$ be a $d$-dimensional Hilbert space equipped with the inner product $\langle \boldsymbol{x}_1, \boldsymbol{x}_2 \rangle := \sum_{i \in [d]} \boldsymbol{x}_1^{(i)} \cdot \boldsymbol{x}_2^{(i)}$ for all $\boldsymbol{x}_1, \boldsymbol{x}_2 \in \mathcal{H}$. In the rest of this paper, we will call $\boldsymbol{x}$ a *patch-like vector* if $\boldsymbol{x} \in \mathcal{H}$.

Real-world images can be split into patches. Motivated by this, our paper uses disjoint patch-like vectors to represent the images. More specifically, we say that $\boldsymbol{z}$ is $P$-*patch data* (or $\boldsymbol{z}$ has $P$-*patch structure*) if $\boldsymbol{z}$ can be represented by an array of patch-like vectors, i.e., $\boldsymbol{z} = (\boldsymbol{x}_1, \boldsymbol{x}_2, \cdots, \boldsymbol{x}_P)^T \in \mathcal{H}^P$. For all $i \in [P]$, we call $\boldsymbol{x}_i$ a $d$-dimensional *patch* of $\boldsymbol{z}$.

Our paper assumes that both the normal data and the anomalies are $P$-patch data with fixed $P$ and $d$. In the context of convolutional neural networks, the (convolutional) kernels $\boldsymbol{w}$ and the patches have identical sizes and shapes. Therefore, we assume that the kernels $\boldsymbol{w}$ are also patch-like vectors. For all $\boldsymbol{x}, \boldsymbol{w} \in \mathcal{H}$, we call $\langle \boldsymbol{x}, \boldsymbol{w} \rangle$ the *convolution* of $\boldsymbol{x}$ and $\boldsymbol{w}$ by convention.

**Features.** We characterize the features as $d$-dimensional orthogonal patch-like vectors. Specifically, we assume that the set of all features $\mathcal{V} := \{\boldsymbol{v}_1, \boldsymbol{v}_2, \cdots, \boldsymbol{v}_d\}$ form an orthonormal basis of $\mathcal{H}$. Based on whether it is related to the semantic information of the normal data, we further categorize the features into normal and auxiliary features. As illustrated in Figure 1, if hand-written digit "1" is considered as the normal data, then the patches $\boldsymbol{x}_{a,1}$, $\boldsymbol{x}_{a,2}$, $\boldsymbol{x}_{b,1}$, and $\boldsymbol{x}_{c,1}$ might contain normal features that capture part of the semantic information of the normal data. In Figure 1.c, $\boldsymbol{x}_{c,2}$ contains an auxiliary feature that captures the semantic information of a semantic anomaly of "7" instead of the normal data "1". However, we argue that these auxiliary features can be regarded as either acceptable errors or sample diversity, and thus might appear in normal data with a small yet non-negligible probability.

In this paper, we formalize the collection of normal and auxiliary features of the normal data as disjoint subsets of $\mathcal{V}$, noted by $\mathcal{V}_{nor}$ and $\mathcal{V}_{aux}$, respectively. Without loss of generality, we assume that $\mathcal{V}_{nor} = \{\boldsymbol{v}_1, \boldsymbol{v}_2\}$ and $\mathcal{V}_{aux} = \{\boldsymbol{v}_3, \boldsymbol{v}_4, \cdots, \boldsymbol{v}_d\}$. Notice that our results can be easily extended to $\mathcal{V}_{nor}$ and $\mathcal{V}_{aux}$ taking arbitrary complementary subsets of $\mathcal{V}$ at the cost of tedious computation.

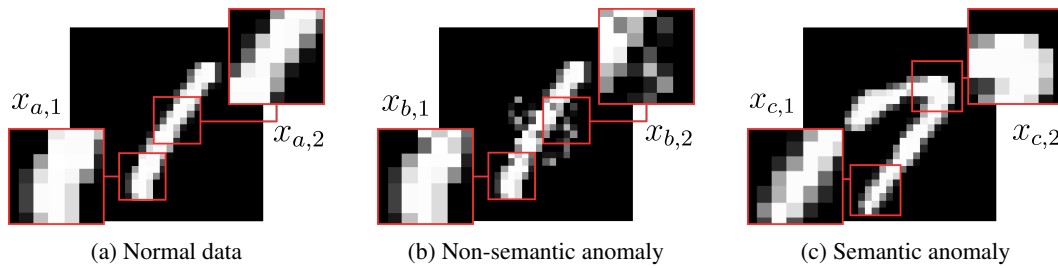

(a) Normal data      (b) Non-semantic anomaly      (c) Semantic anomaly

Figure 1: **Patches, features, and different types of anomalies**. As an illustrative example, consider the handwritten digits with only "1" being normal data. Patches $\boldsymbol{x}$ are small parts of the images, e.g., those outlined with red boxes. $\boldsymbol{x}_{a,1}$, $\boldsymbol{x}_{a,2}$, $\boldsymbol{x}_{b,1}$, and $\boldsymbol{x}_{c,1}$ contain a **normal feature** that captures the semantic information of "1". Figure 1b is called the non-semantic (or sensory) anomaly in the literature since $\boldsymbol{x}_{b,2}$ contains **non-feature noise**. $\boldsymbol{x}_{c,2}$ contains an **auxiliary feature** that does not capture the semantic information of "1", making Figure 1c a semantic anomaly.

## 2.1 DATA DISTRIBUTION

Our paper assumes that the normal data $z$ is drawn independently from an implicit distribution $D_{nor}$. Given normal data $z$, we assume that each of the patches of $z$ contains one feature. Here, we say that a patch $x_i$ *contains* a feature $v_k$ if $x_i = v_k + \rho$, in which $\rho$ is a noise term that satisfies $|\langle \rho, v_k \rangle| \leq \varepsilon$ for some given constant $\varepsilon > 0$.

For any $k \in [d]$ and $z \sim \mathcal{D}_{nor}$, we let the probability of the event $\{\exists x_i \in z \text{ such that } x_i \text{ contains } v_k\}$ be a given constant $\beta_k \in [0, 1]$. Intuitively, normal data would contain more normal features than auxiliary features. For simplicity, we assume that $\beta_k = 1$ for all $k \in \{1, 2\}$, i.e., normal data surely contain normal features. As for the auxiliary features, we require $\beta_k < 1$ for any $k \in \{3, 4, \cdots, d\}$ and $\sum_{k'=1}^{d} \beta_{k'} = P$ to ensure there is no empty patches. In summary, we adopt the following assumption for the normal data.

**Assumption 2.1** (Normal data model). We assume that normal data $z = (x_1, x_2, \cdots, x_P)$ is drawn independently from an implicit distribution $D_{nor}$ that satisfies

1. for all $i \in [P]$, $\exists k \in [d]$ such that $x_i$ contains $v_k$;

2. for all $k \in [d]$, $\mathbf{P}_{z \sim \mathcal{D}_{nor}} [\exists i \in [P] \text{ s.t. } x_i \text{ contains } v_k] = \beta_k$;

3. $\beta_1 = \beta_2 = 1$, $\beta_k < 1$ for all $k \in \{3, 4, \cdots, d\}$, and $\sum_{k'=1}^{d} \beta_{k'} = P$.

Given $N \in \mathbb{Z}^+$, let $[z_N]$ be the training data that is drawn independently from $D_{nor}$, in which $z_n \in [z_N]$ is noted by $z_n := (x_{n,1}, x_{n,2}, \cdots, x_{n,d})$ in the remainder of this paper. Notably, RBAD does not require anomalies during training, which constitutes one of the core advantages of this method. We will introduce the data model for the anomalies later in Section 3.2.

## 2.2 NETWORK STRUCTURE

Let $\phi : \mathcal{H}^P \to \mathcal{H}^P$ be a function that reconstructs the inputs. It is not hard to show that simple models (e.g., principal component analysis) would fail to reconstruct the normal data due to the unspecified ordering rule of the patches. Therefore, we use a slightly more sophisticated model, i.e., the 2-layer convolutional autoencoder.

An autoencoder $\phi$ is composed of an encoder $\phi_e$ and a decoder $\phi_d$, i.e., $\phi(z) = \phi_d(\phi_e(z))$ for all $z \in \mathcal{H}^P$. Denote the number of channels by $C = c_2 d$ for some given constant $c_2$ and let $w = (w_1, w_2, \cdots, w_C)$ be the weight of $\phi_e$, in which $w_j$ ($j \in [C]$) are kernels as described earlier. Given input $z \in \mathcal{Z}$, the output of $\phi_e$ is noted as $\phi_e(z; w) = (\phi_e(z; w_1), \cdots, \phi_e(z; w_C))$, in which

$$\phi_e(z; w_j) = (\sigma(\langle w_j, x_1 \rangle), \cdots, \sigma(\langle w_j, x_P \rangle)). \tag{1}$$

Here, $\sigma(\cdot)$ is the *smoothed ReLU function* defined as

$$\sigma(x) = \begin{cases} 0 & \text{if } x \leq 0; \\ (2c_0)^{-1} x^2 & \text{if } 0 < x \leq c_0; \\ x & \text{if } x > c_0, \end{cases} \tag{2}$$

for given constant $c_0$. Unlike the original ReLU function, $\sigma$ is derivable near 0. Note that $\phi_e(z; w_j)$ is also $P$-patch data, but the dimension of the patch is reduced from $d$ to 1.

The decoder reconstructs $z$ according to the features learned by $\phi_e$. To illustrate how the features are used in construction, consider the following ideal case in which $w$ perfectly learns the features, i.e., let $C(d) = d$ and $w_j = v_j$ for all $j \in [d]$. In this case, we can write

$$\tilde{x}_i = \sum_{j \in [d]} \langle x_i, w_j \rangle w_j = x_i, \tag{3}$$

which provides a trivial yet perfect reconstruction of the patches in $z$.

We modify the reconstruction in Equation (3) by considering an over-parameterized setting with a max pooling operation. We make these modifications for the following reasons. First of all, the proof of Theorem 1 implies that the reconstruction in Equation (3) would easily converge to local

minima; we refer the readers to Remark E.1 for detailed discussion. To avoid reaching local minima, we consider an over-parameterized setting with $c_2 > 1$.

In practice, the max pooling operation is placed after the convolution layer to select the most representative patch, i.e., the patch with the maximum convolution value. For ease of computation, our work also adds a max pooling operation in the decoder. For all $j \in [C]$, let $\delta(i, j)$ be a binary-value function that takes 1 when

$$i = \arg\max_{i' \in [P]} \sigma(\langle \boldsymbol{w}_j, \boldsymbol{x}_{i'} \rangle) \tag{4}$$

and 0 otherwise. Finally, we have the over-parameterized version of Equation (3) with max pooling

$$\phi_d^{(i)}(\phi_e(\boldsymbol{z})) = \sum_{j \in [C]} \sigma(\langle \boldsymbol{w}_j, \boldsymbol{x}_i \rangle) \delta(i, j) \boldsymbol{w}_j. \tag{5}$$

Intuitively, for all $\boldsymbol{x}_i \in [\boldsymbol{x}_P]$, $\phi_d^{(i)}(\phi_e(\boldsymbol{z}))$ reconstructs $\boldsymbol{x}_i$ by adding together those $\boldsymbol{w}_j$ such that $\boldsymbol{x}_i$ is the "most related" patch to $\boldsymbol{w}_j$. Combining Equations (4) and (5), our decoder $\phi_d$ is defined as

$$\phi_d(\phi_e(\boldsymbol{z})) = \left( \phi_d^{(1)}(\phi_e(\boldsymbol{z})), \cdots, \phi_d^{(P)}(\phi_e(\boldsymbol{z})) \right). \tag{6}$$

In the rest of this paper, we will always omit the parameter $\boldsymbol{w}$ if it does not cause confusion.

## 2.3 TRAINING PARADIGM

We train the autoencoder $\phi$ under a standard empirical risk minimization paradigm (Shalev-Shwartz & Ben-David, 2014). As mentioned in Section 2.1, given $N \in \mathbb{Z}^+$, the training data $[\boldsymbol{z}_N]$ are drawn independently from distribution $D_{nor}$. To measure how well $\phi$ reconstructs the input data, we let the *reconstruction loss* be

$$\ell(\boldsymbol{z}; \boldsymbol{w}) = \|\phi(\boldsymbol{z}; \boldsymbol{w}) - \boldsymbol{z}\|^2. \tag{7}$$

The corresponding *(empirical) reconstruction error* is defined as $\boldsymbol{R}_N(\boldsymbol{w}) = \frac{1}{N} \sum_{n \in [N]} \ell(\boldsymbol{z}_n; \boldsymbol{w})$. We minimize $\boldsymbol{R}_N(\boldsymbol{w})$ using gradient descent with random initialization. For all $j \in [C]$ and $t > 0$, we iterate $\boldsymbol{w}_j$ by

$$\boldsymbol{w}_j^{t+1} = \boldsymbol{w}_j^t - \eta_t \nabla_{\boldsymbol{w}_j} \boldsymbol{R}_N(\boldsymbol{w}^t) \tag{8}$$

for all $t \in \mathbb{N}$. The learning rate $\eta_t$, specified in Appendix D, is a parameter that gradually decreases to 0 as the iteration step $t$ grows. Note that $\eta_t$ is independent of $\boldsymbol{w}_j$. For all $j \in [C]$, the random initialization of $\boldsymbol{w}_j$ is given by $\boldsymbol{w}_j^0 \sim \mathcal{N}(0, \sigma_0 \boldsymbol{I})$, in which $\mathcal{N}(0, \sigma_0 \boldsymbol{I})$ is the $d$-dimensional Gaussian distribution and $\sigma_0$ is a given constant.

## 3 THEORETICAL ANALYSIS

Our analyses can be roughly divided into two phases: the *training phase* (Section 3.1) and the *reconstruction phase* (Section 3.2). The main results and intuitions are summarized in the following theoretical roadmap. We include a list of summarized terms and notations in Appendix C. The proofs are postponed to Appendix E.

## 3.1 TRAINING PHASE

We start with analyzing the growth of the kernels. Given $n \in [N]$ and $i \in [P]$, we use

$$\Delta_{n,i} = \boldsymbol{x}_{n,i} - \phi_d^{(i)}(\phi_e(\boldsymbol{z}_n)) \tag{9}$$

to denote the difference between the input patch $\boldsymbol{x}_{n,i}$ and its reconstruction $\phi_d^{(i)}(\phi_e(\boldsymbol{z}_n))$. Recall that the growth of the kernel is given by $\boldsymbol{w}_j^{t+1} - \boldsymbol{w}_j^t = -\eta_t \nabla_{\boldsymbol{w}_j} \boldsymbol{R}_N(\boldsymbol{w}^t)$ in Equation (8), in which $\nabla_{\boldsymbol{w}_j} \boldsymbol{R}_N(\boldsymbol{w}^t)$ can be calculated by the following proposition.

**Proposition 3.1.** *For any $j \in [C]$, the gradient of the empirical reconstruction error is*

$$\nabla_{\boldsymbol{w}_j} \boldsymbol{R}_N(\boldsymbol{w}) = -\frac{2}{N} \sum_{n,i} \delta_n(i, j) \left[ \sigma(\langle \boldsymbol{w}_j, \boldsymbol{x}_{n,i} \rangle) \Delta_{n,i} + \langle \Delta_{n,i}, \boldsymbol{w}_j \rangle \sigma'(\langle \boldsymbol{w}_j, \boldsymbol{x}_{n,i} \rangle) \boldsymbol{x}_{n,i} \right]. \tag{10}$$

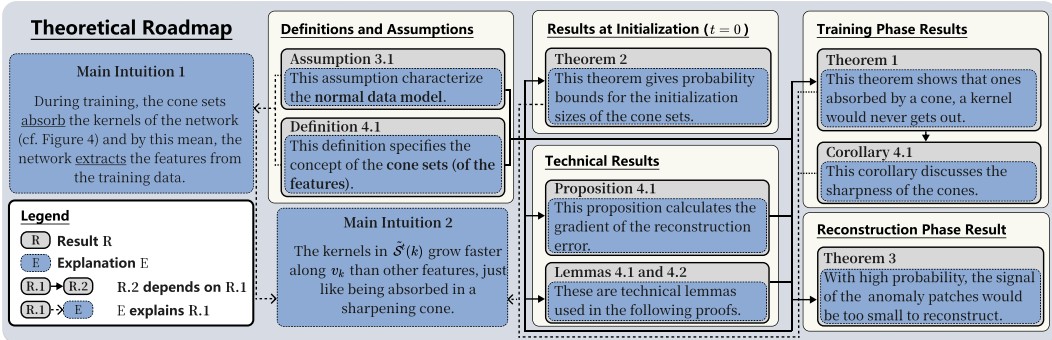

Figure 2: A *theoretical roadmap* of our analysis. We visualize the dependency of the results and attach a brief explanation for each. We also summarize two major intuitions.

Proposition 3.1 implies a positive correlation between the growth rate of $\boldsymbol{w}_j$ and the size of $\boldsymbol{w}_j$ (i.e., $\|\boldsymbol{w}_j\|$), considering that the sizes of $\sigma(\langle \boldsymbol{w}_j, \boldsymbol{x}_{n,i}\rangle)$, $\langle \Delta_{n,i}, \boldsymbol{w}_j\rangle$, and $\sigma'(\langle \boldsymbol{w}_j, \boldsymbol{x}_{n,i}\rangle)$ in Equation (10) are all directly related to $\|\boldsymbol{w}_j\|$. The following lemma further controls the initial value and the growth rate of $\|\boldsymbol{w}_j\|$.

**Lemma 3.1.** *Let $p_0 = p_0(d)$ and $f_h = f_h(d)$ be parameters. For all $j \in [C]$, we have*

$$\|\boldsymbol{w}_j^0\| \leq \sigma_0 f_h \tag{11}$$

*with probability at least $1 - p_0$ with regard to the random initialization of $\boldsymbol{w}$. Moreover, for all $t \in [T]$, the size of $\boldsymbol{w}_j^t$ is controlled by*

$$\|\boldsymbol{w}_j^t\| \leq t\sigma_0 f_h. \tag{12}$$

*with probability at least $1 - p_0$. Here, $T(t, d)$ is a parameter depending on $t$ and $d$.*

*Remark* 3.1 (**Parameter Analysis**). Some parameters in this paper, e.g., $p_0$, $f_h$, and $T$, have clear geometrical or probabilistic intuition. We will explain them in later parts of this paper. A summary for the parameters can be found in Appendix C. For simplicity, we will omit the sup- or sub-scripts $t$ and $d$ if it does not cause confusion. A set of recommended parameters is given in Appendix D. ▲

Note that the probability lower bound for Equations (11) and (12) to hold (i.e., at least $1 - p_0$) is the same because, after random initialization, the kernels would grow in a deterministic manner; the iteration rule given by Equation (8) introduces no randomness. The above discussion implies that those kernels with larger initial values will grow faster. A simple deduction is that after multiple iterations, the *faster-growing kernels* would take larger values, thus significantly affecting the reconstruction phase of RBAD.

The following definition introduces the main intuition of our analysis.

**Definition 3.1** (Cone set). Let $c_1 = c_1(d) \in (0, 1)$ and $f_r = f_r(d, t) \leq c_1(d)f_h(d)$ be parameters with $f_r = f_h \cdot o(t)$. For any $t \in [T]$ and $k \in [d]$, let the cone set of feature $\boldsymbol{v}_k$ at iteration $t$, denoted by $\tilde{\mathcal{S}}^t(k)$, be the collection of $\boldsymbol{w}_j \in [w_C]$ such that

- $\langle \boldsymbol{w}_j^t, \boldsymbol{v}_k \rangle \geq t\sigma_0 c_1 f_h$, and

- for any $k' \neq k$, $\langle \boldsymbol{w}_j^t, \boldsymbol{v}_{k'} \rangle \leq t\sigma_0 f_r$.

We say that kernel $\boldsymbol{w}_j$ is **absorbed** by the cone sets of feature $\boldsymbol{v}_k$ if $\exists t_0 \in [T]$ such that $\boldsymbol{w}_j^t \in \tilde{\mathcal{S}}^t(k)$ for all $t \geq t_0$, i.e., after the iteration step $t_0$, the kernel $\boldsymbol{w}_j$ is always in the cone set of feature $\boldsymbol{v}_k$. ▲

*Remark* 3.2 (**Cone Set Intuition**). For any $k \in [d]$ and $t \in [T]$, $k$, $t$ together with the parameters $\sigma_0$, $c_1$, $f_h$ and $f_r$ specify a $d$-dimensional hypercone with height $t\sigma_0 c_1 f_h$ and radius $t\sigma_0 f_r$. According to Definition 3.1, those $\boldsymbol{w}_j^t \in \tilde{\mathcal{S}}^t(k)$ are bounded by the surface of the cone. Since we let $f_r = f_h \cdot o(t)$, the cone would get sharper during training (cf. Figure 3). As a result, the kernels in the cone would get more aligned to feature $\boldsymbol{v}_k$. In particular, the kernels absorbed by the cone set of a feature will eventually approximate to the direction of that feature. ▲

We can obtain the following lemma directly from Definition 3.1.

**Lemma 3.2.** *For all $t \geq 1$, if $j \in \tilde{\mathcal{S}}^t(k)$ and $\boldsymbol{x}_i$ contain $\boldsymbol{v}_k$, then we have $\delta(i, j) = 1$.*

Lemma 3.2 implies that the kernels in the cones of a feature $\boldsymbol{v}_k$ will contribute to the reconstruction of the patches that contain $\boldsymbol{v}_k$ at all iterations. Lemma 3.2 also simplifies the iteration analysis for the kernels in the cones. We can prove the following theorem with the help of Lemmas 3.1 and 3.2.

**Theorem 1.** *For all $k \in [d]$, if $|\tilde{\mathcal{S}}^0(k)| \geq 1$, then $\tilde{\mathcal{S}}^t(k) \subseteq \tilde{\mathcal{S}}^{t+1}(k)$ for all $t \in [T]$.*

*Remark 3.3 (**Interpretation of Theorem 1**).* Theorem 1 characterizes how the kernels of the autoencoder extract features from the training data, providing an answer for question $\boldsymbol{Q}_1$. For all $k \in [d]$ and $t \in [T]$, Theorem 1 implies that $\boldsymbol{w}_j \in \tilde{\mathcal{S}}^t(k)$ will keep growing faster along $\boldsymbol{v}_k$ than along $\boldsymbol{v}_{k'}$ for all $k' \neq k$. The geometrical intuition is that *the kernel will be absorbed by the cone set of a feature if it comes into the cone set once*. After multiple iterations, $\langle \boldsymbol{w}_j^t, \boldsymbol{v}_{k'} \rangle$ would become almost negligible compared to $\langle \boldsymbol{w}_j^t, \boldsymbol{v}_k \rangle$ since $f_r = f_h \cdot o(t)$. The direction of $\boldsymbol{w}_j$ gradually approximates to the direction of $\boldsymbol{v}_k$ as $t$ increases, i.e., $\boldsymbol{w}_j$ learns the direction of $\boldsymbol{v}_k$.

For any $k \in [d]$, the following theorem proves that $\tilde{\mathcal{S}}(k)^0$ is not empty with high probability.

**Theorem 2.** *Let $p_1 = p_1(d)$ and $p_2 = p_2(d)$ be parameters. For all $k \in [d]$, the size of $\tilde{\mathcal{S}}^0(k)$ can be upper-bounded by*

$$|\tilde{\mathcal{S}}^0(k)| \leq Cp_2 + \lambda, \tag{13}$$

*with probability at least $1 - \exp\left(-\frac{3\lambda^2}{6Cp_2 + 2\lambda}\right)$ over random initialization. The lower-bound*

$$|\tilde{\mathcal{S}}^0(k)| \geq Cp_1 - \lambda \tag{14}$$

*hold with probability at least $1 - \exp\left(-\frac{\lambda^2}{2Cp_2}\right)$.*

Theorems 1 and 2 together characterize the growth of the kernels $\boldsymbol{w}_j$ in the cones of the features. Figure 3 demonstrates the geometric intuition of our results.

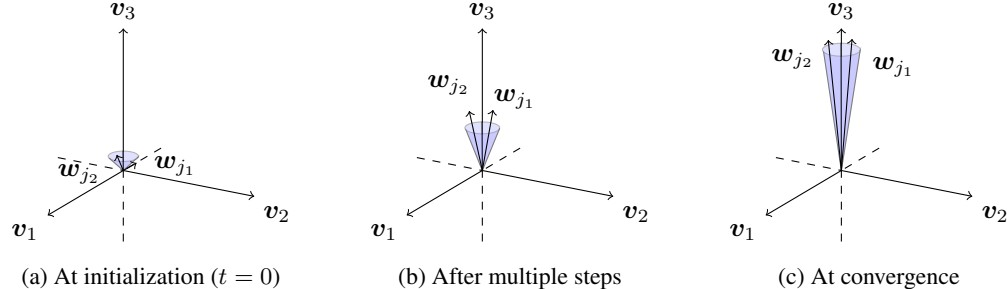

(a) At initialization ($t = 0$)    (b) After multiple steps    (c) At convergence

Figure 3: **The geometry intuition for Theorems 1 and 2.** Consider $d = 3$ and $\boldsymbol{w}_{j_1}, \boldsymbol{w}_{j_2} \in \tilde{\mathcal{S}}(3)^0$. The kernels are absorbed in the cone represented by the light blue surface. Definition 3.1 implies that the cone would get sharper as $t$ increases. Theorem 1 proves that $\boldsymbol{w}_{j_1}, \boldsymbol{w}_{j_2}$ will be absorbed if $\boldsymbol{w}_{j_1}, \boldsymbol{w}_{j_2} \in \tilde{\mathcal{S}}(3)^0$. Theorem 2 implies that the cone set is non-empty with high probability.

It is worth mentioning that the above results hold for all feature $\boldsymbol{v}_k$ with $\beta_k \neq 0$, regardless of whether $\boldsymbol{v}_k$ is a normal feature or an auxiliary feature. Recall that we use auxiliary features to characterize patterns not directly related to the semantics of the normal data. As a minor implication, we prove that neural networks would also learn the auxiliary features. The following corollary discusses the tightness of the bounds for different features.

**Corollary 3.1** (Informal). *Recall that Definition 3.1 implies a growth rate gap between the kernels growth along the "cone feature's direction" and other features direction. For those kernels absorbed in the cones of normal features, the growth rate gap is larger than those in auxiliary features.*

The formal version of this corollary is postponed to Appendix E due to limited space.

In summary, this section analyzes the training phase of RBADs. Notice that the question $\boldsymbol{Q}_1$ is answered in this section. It remains to discuss how the features are used in the reconstruction phase and why the anomalies are hard to reconstruct.

## 3.2 RECONSTRUCTION PHASE

Since the learning rate $\eta_t$ decreases to $0$ as $t$ grows, the kernels would gradually converge. Denote the converged kernels by $\boldsymbol{w}^* := (\boldsymbol{w}_1^*, \boldsymbol{w}_2^*, \cdots, \boldsymbol{w}_P^*)$ and the corresponding network by $\phi^*$. Section 3.2 discusses how $\phi^*$ reconstruct the input images.

We start by introducing the data model for anomalies. Let $\boldsymbol{z} \sim D_{nor}$ be normal data (cf. Assumption 2.1) and $\boldsymbol{x}_a \in \mathcal{H}$ be a patch. Given $i_0 \in [P]$, the *anomaly* is defined as

$$\boldsymbol{z}(i_0, \boldsymbol{x}_a) := (\boldsymbol{x}_1, \cdots, \boldsymbol{x}_{i_0-1}, \boldsymbol{x}_a, \boldsymbol{x}_{i_0+1}, \cdots, \boldsymbol{x}_P), \tag{15}$$

i.e., one of the patches in $\boldsymbol{z}$ is replaced by $\boldsymbol{x}_a$. We call $\boldsymbol{z}(i_0, \boldsymbol{x}_a)$ a *semantic anomaly* if $\exists k \in [d]$ such that $\boldsymbol{x}_a$ contains $\boldsymbol{v}_k$ and a *non-semantic anomaly* if otherwise. For non-semantic anomalies, we further assume that $\boldsymbol{x}_a$ is drawn from the uniform distribution supported on $\{\boldsymbol{x} : \|\boldsymbol{x}\| = 1\}$. Note that our analysis can be trivially extended to the case where multiple patches are replaced.

In order to answer $\boldsymbol{Q}_2$, we go on to analyze the behavior of the kernels in the reconstruction phase. Taking $\boldsymbol{z}(i, \boldsymbol{x}_a)$ as the input of $\phi^*$, by definition, we have

$$\phi_e^{(i)}(\boldsymbol{z}(i_0, \boldsymbol{x}_a); \boldsymbol{w}_j^*) = \sigma(\langle \boldsymbol{w}_j, \boldsymbol{x}_i \rangle) = \phi_e^{(i)}(\boldsymbol{z}; \boldsymbol{w}_j^*) \tag{16}$$

for all $i \in [P]$ such that $i \neq i_0$. Since $\mathcal{V}$ is an orthonormal basis of $\mathcal{H}$, we can decompose $\boldsymbol{x}_a$ as

$$\boldsymbol{x}_a = \sum_{k \in [d]} \langle \boldsymbol{x}_a, \boldsymbol{v}_k \rangle \, \boldsymbol{v}_k. \tag{17}$$

By taking Equation (17) into Equation (1), the replaced patch is encoded as follows.

$$\phi_e^{(i_o)}(\boldsymbol{z}(i_0, \boldsymbol{x}_a); \boldsymbol{w}_j^*) = \sigma(\langle \boldsymbol{w}_j^*, \sum_{k \in [d]} \langle \boldsymbol{x}_a, \boldsymbol{v}_k \rangle \, \boldsymbol{v}_k \rangle) = \sigma(\sum_{k \in d} \langle \boldsymbol{x}_a, \boldsymbol{v}_k \rangle \langle \boldsymbol{w}_j^*, \boldsymbol{v}_k \rangle). \tag{18}$$

For all $j \in [C]$, by definition, the max pooling operation only reserves $\max_{i \in [P]} \sigma(\langle \boldsymbol{w}_j, \boldsymbol{x}_i \rangle)$. We have proved in Section 3.1 that the direction of the kernels would gradually get closer to the directions of the features. For non-semantic anomalies, we can prove the following theorem.

**Theorem 3.** *For all $\theta \in (0,1)$, $\forall j \in [C]$, and $\forall k \in [d]$, if $\langle \boldsymbol{w}_j, \boldsymbol{v}_k \rangle > (1 - \theta) \cdot \|\boldsymbol{w}_j\|$, then*

$$\langle \boldsymbol{w}_j, \boldsymbol{x}_a \rangle < \|\boldsymbol{w}_j\| \cdot \max\{2\theta, \frac{1}{\theta\sqrt{d-1}}\} \tag{19}$$

*hold with probability at least $1 - 2\theta \cdot \exp(-1/(2\theta^2))$.*

Theorem 3 implies that the convolution between non-semantic anomalies $\boldsymbol{x}_a$ and the kernels in the cones will take small values with high probability. Therefore, the signal in the replaced patch will be filtered by the max pooling operation, making the replaced patches in the anomalies hard to reconstruct with high probability. The above discussion answers the questions $\boldsymbol{Q}_2$ and $\boldsymbol{Q}_3$.

*Remark* 3.4 (***Semantic Anomalies are hard to detect***). As reported in the previous survey (Ruff et al., 2021), RBADs are unsuitable for detecting semantic anomalies. As discussed in Allen-Zhu & Li (2023), data with different semantic information could contain similar features. In our model, denote the feature contained in the replaced patch of a semantic anomaly by $\boldsymbol{v}_{k_0}$. Lemma 3.2, Theorem 1, and Theorem 2 together imply that the autoencoder would also learn the semantic information of $\boldsymbol{v}_{k_0}$ if $\beta_{k_0} \neq 0$. As a result, the kernels in the cone of $\boldsymbol{v}_{k_0}$ would align larger value (depending on $\beta_{k_0}$) to $\langle \boldsymbol{w}_j, \boldsymbol{x}_a \rangle$ for semantic anomalies than non-semantic anomalies. The reconstruction error or the semantic anomalies would be smaller than the non-semantic anomalies, making the semantic anomalies harder to detect.

In summary, this section analyzes the reconstruction phase of the obtained autoencoder and explains why the non-semantic anomalies are hard to reconstruct. We also discuss why RBADs are unsuitable for detecting semantic anomalies. All the questions raised in Section 1 are answered.

## 4 NUMERICAL VALIDATION

This section presents synthesized and real-world experiments that validate our theoretical findings. Due to limited space, the implementation details and auxiliary results are reported in Appendix G.

**Synthesized Experiments**   We conduct this experiment on a synthesis dataset that contains $4000$ training data and $1000$ test data for each of the normal data, semantic anomalies, and non-semantic anomalies. The training and test data are drawn from the distribution specified by Assumption 2.1. We consider different noise levels from $\varepsilon \in \{0.1, 0.01, 0.001\}$. The number of patches $P$ is selected from $\{20+4u : u = 0, 1, \cdots, 10\}$ and we let $d = \text{int}(c_3 P)$ for $c_3 \in \{1.2, 1.5, 2\}$. We explore other choices of $d$ and $C$ in Appendix G.2. The anomalies are generated according to Equation (15). We consider an over-parameterized scenario with $C = \text{int}(1.2d)$. The network is specified in Section 2. We use the mean square error (MSE) for the loss function and SGD for optimization.

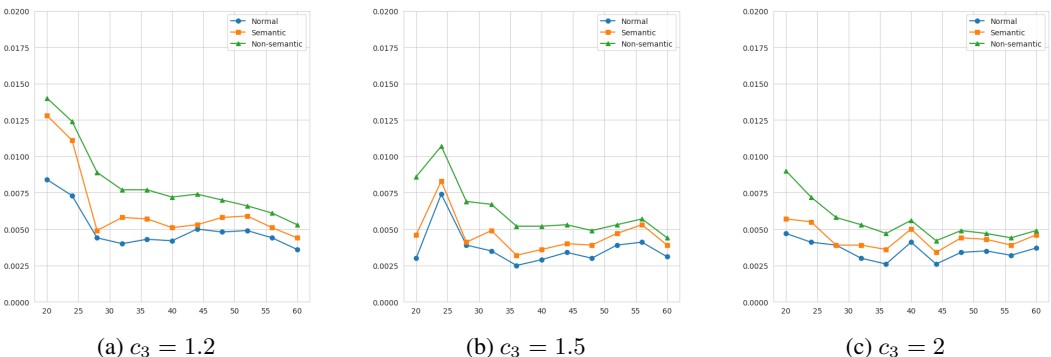

(a) $c_3 = 1.2$          (b) $c_3 = 1.5$          (c) $c_3 = 2$

Figure 4: **Results of the synthesized experiments**. In these figures, the $x$-axis is the number of patches $P$. The $y$-axis reports the reconstruction losses. We let $d = \text{int}(c_3 P)$ for $c_3 \in \{1.2, 1.5, 2\}$ and $C = \text{int}(1.2d)$ in the study. All of the experiments exhibit clear gaps between the reconstruction errors of the normal data and the non-semantic anomalies. Besides, as discussed in Remark 3.4, the gap between the reconstruction errors of the normal data and the semantic anomalies is not as significant as that between the normal data and the non-semantic anomalies.

**Visualization of the kernels**   Figure 5 visualize the first-layer features learned from MNIST and CIFAR-10, which validates our proposed cone set intuition. It can be observed that the shapes and colors of the kernels do not significantly change during training because, as claimed in Section 3, the kernels are absorbed by the cones of the features. Notice that some of the kernels remain noisy even after 50 epochs. We interpret these kernels as those $\boldsymbol{w}_j$ such that $j \notin \cup_{k \in [d]} \tilde{\mathcal{S}}(k)$.

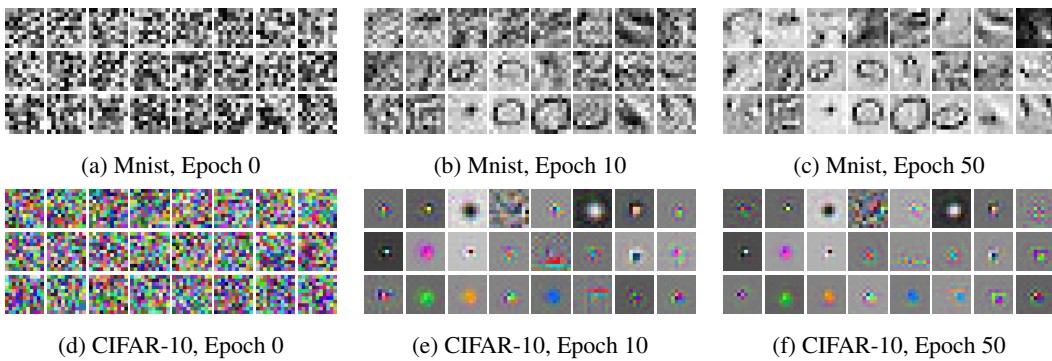

(a) Mnist, Epoch 0          (b) Mnist, Epoch 10          (c) Mnist, Epoch 50

(d) CIFAR-10, Epoch 0          (e) CIFAR-10, Epoch 10          (f) CIFAR-10, Epoch 50

Figure 5: Visualization of the kernels in the first layer of the autoencoder.

## 4.1 DISCUSSIONS

We discuss the significance of the experiments as follows.

**Synthetic experiments (Figure 4)**   : The synthetic experiments are designed to verify the theoretical modeling and analysis in Section 3.2. In learning theory community, it is a common practice to conduct synthetic experiments under the same setting as the theoretical analysis, serving to

demonstrate whether the theoretical modeling can effectively characterize the practical problems (i.e., RBAD in our paper). In the caption of Figure 4, we briefly explain how the experimental results consist with our theory and the existing results. More specifically, the synthetic experiments conducted under our theoretical setting exhibit clear reconstruction error gaps between normal data and the anomalies, which is a key rationale underlying RBAD.

**Cone set intuition (Figure 5)** : We use experiments on real-world datasets to illustrate the proposed cone set intuition. As a supplement to the existing explanations, the experiments in Figure 5 discusses the manifestations of the cone set intuition on networks trained with MNIST and CIFAR-10, which indirectly verifies the validity of our theory.

Take the results on MNIST (Figures 5.(a)-5.(c)) as an example. The visualized kernels are randomly initialized at epoch 0. It can be observed by Figure 5.(a) that the kernels are completely noisy. By comparing Figures 5.(b) and 5.(c), the following two observations are closely related to our proposed theory.

1. Some kernels (e.g., (row-2,col-3) and (row-3, col-4)) have exhibit clear contours at the early stage of training (epoch 10) and do not undergo significant changes at convergence (epoch 50).

2. Meanwhile, some other kernels (e.g., (row-1, col-1) and (row-3, col-2)) fail to exhibit distinct outlines even after convergence, which aligns with the case when the kernel is not absorbed by any of the cones.

Both of the above points are consistent with the probability-based training dynamics proposed in our work. As a supplement to the existing explanations, these experiments discusses the manifestations of the cone set intuition on networks trained with MNIST and CIFAR-10, which indirectly verifies the validity of our theory.

## 5 CONCLUSION

This paper studies the training dynamics of a 2-layer convolutional autoencoder and how it reconstructs different types of inputs. Particularly, we answer $Q_1$ - $Q_3$ raised in Section 1 as follows.

($Q_1$) How does the model learn the features from training data in RBAD? ($A_1$) *The kernels absorbed by the cone sets of the features would approximate the features.*

($Q_2$) What is the role of the learned features in the reconstruction phase? ($A_2$) *The features process the signal from the input patches and use them for reconstruction.*

($Q_3$) Why are the anomalies harder to reconstruct? ($A_3$) *The processed signal from the auxiliary features and the random noise is smaller than that from the normal features.*

Our theoretical findings and intuitions are supported by synthesized and real-world experiments.

### ETICS STATEMENT

All authors of this submission fully acknowledge and strictly adhere to the ICLR Code of Ethics. Prior to submitting this paper, every author has carefully read and understood the requirements of the Code of Ethics, and commits to complying with all relevant provisions throughout the entire process of paper submission, review, and subsequent discussion.

This research does not involve human subjects, nor does it include the release of any datasets that may pose risks to privacy or security. In the design of research methodologies and the discussion of potential applications, the authors have fully considered issues related to fairness, bias, and research integrity, and have taken necessary measures to avoid any potential harm. Regarding conflicts of interest and sponsorship, all authors confirm that there are no undisclosed financial or non-financial conflicts of interest. For matters related to legal compliance and research ethics, the research strictly follows the relevant laws, regulations, and academic norms of the research field.

## REPRODUCIBILITY STATEMENT

The main contribution of this paper is establishing a theoretical framework for analyzing RBAD and providing a plausible explanation for RBAD. Complete proof can be found in the Appendix E. Clear explanations and discussions are provided just after every theorem and corollary.

### ACKNOWLEDGMENTS

This work is supported by the Key R&D Program of Hubei Province under Grant 2024BAB038, the National Key R&D Program of China under Grant 2023YFC3604702.

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

## A RELATED WORKS

The present paper studies the theoretical properties of RBAD through the lens of feature learning. The related works of RBADs and feature learning are discussed as follows.

## A.1 ANOMALY DETECTION AND RBAD

In contrast to many machine learning topics, anomaly detection (AD) Ma et al. (2025b) adopts the "open-world" assumption in which the training and test data are not drawn from the same distribution. According to how the test distribution differs from the training distribution, AD can be further categorized into semantic AD (Ahmed & Courville, 2020; Deecke et al., 2021) and non-semantic AD (Bergmann et al., 2021; Mousakhan et al., 2023). In this work, we characterize the normal data and both types of anomalies using a unified probabilistic model.

The "open-world" setting of AD is also adopted in some closely related topics, e.g., outlier detection (OD) (Wang et al., 2019), novelty detection (ND) (Pimentel et al., 2014), and out-of-distribution (OOD) detection (Graham et al., 2023; Ma et al., 2024; 2026; 2025a). Yang et al. (2021) review the similarities and the subtle differences of these topics. Although AD, OD, ND, and OOD detection are different topics, these terms (especially AD, OD, and ND) are used interchangeably in some previous works (Ruff et al., 2021; Pang et al., 2022). It is worth mentioning that reconstruction-based methods can also be used in ND (Pidhorskyi et al., 2018; Perera et al., 2019) and OOD detection (Zhou, 2022). The analysis in this paper can be easily extended to the reconstruction-based methods for ND and OOD detection.

## A.2 FEATURE LEARNING

In this paper, the term "feature learning" refers to the theoretical paradigm proposed by Allen-Zhu & Li (2021). Compared to other theoretical paradigms, e.g., the uniform convergence paradigm (Wainwright, 2019; Shalev-Shwartz & Ben-David, 2014; Vapnik, 2000) and the convergence analysis paradigm (Archibald et al., 2022; Shalev-Shwartz & Ben-David, 2014), feature learning provides a more delicate discussion on how the weights of the neural networks iterates.

As the first work of feature learning, Allen-Zhu & Li (2021) analyzes how a 2-layer ReLU network extracts features from the training data. Following Allen-Zhu & Li (2021), a series of works use the tools from feature learning to study different topics, including knowledge distillation (Allen-Zhu & Li, 2023), contrastive learning (Wen & Li, 2021), and neural collapse Cao et al. (2025). In this work, we study RBAD under this paradigm by analyzing how 2-layer convolutional autoencoders extract features from training data and reconstruct the samples. To the best of our knowledge, this paper is the first to analyze generative models using feature learning.

## B LIMITATIONS

This section further discusses the theoretical assumptions and the limitations of this paper.

In this paper, we introduce the Hilbert space $\mathcal{H}$ of patch-like vectors and assume the data have $P$-patch structure. The patches are essentially "sub-matrices" or "sub-tensors" of the original image. Previous works have characterized the patches as vectors since the operations (e.g., addition, scalar multiplication, and convolution) of the patches satisfy the axioms of the Hilbert space when the patches are non-overlapping. However, the patches are not equivalent to vectors in general; to our knowledge, existing structures in vector space cannot characterize the overlapping patches.

In practice, the max pooling operation is placed after the convolution layer, calculating the maximum convolution outputs of several overlapping patches close to each other and possibly containing the same feature. In this sense, the patch with the maximum convolution value might contain the most representative version of the feature. Max pooling would greatly reduce the output dimension while not significantly affecting the value since the reserved value acts as a representative of the others.

In this paper, we simplify the overlapping operation and the relationship between the patches and the features. We use non-overlapping patches to replace the most representative overlapping patches. Besides, the max pooling operation is further simplified in our paper to calculate the maximum value for the convolution of all the patches instead of those close to each other. As a result, according to Equation (10), the growth of the kernels can be decomposed to the combination of two directions.

The max pooling operation in our paper significantly simplifies the analysis process. Without max pooling, the growth of $\boldsymbol{w}_j$ ($j \in [C]$) would be written as the weighted sum of $\Delta_{n,i}$ and $\boldsymbol{x}_{n,i}$ for all $i \in [P]$. However, ignoring the growth of $\boldsymbol{w}_j$ along minor directions would cause problems, e.g., the

zero reconstruction phenomena mentioned in Section 3.2. Nevertheless, the kernel will still grow faster along the direction of the most representative patch since max pooling reserves the maximum convolution value. The max pooling operation does not change the intrinsic quality of the problem.

In summary, we argue that the $P$-patch data structure and the max pooling opeation are coupled with each other and should be adopted entirely. The integrated model could provide a simple yet representative characterization of the practical problem.

## C  TERMS AND NOTATIONS

We include a list of terms and notations in this section.

- Bold letters (e.g., $\boldsymbol{x}$, $\boldsymbol{v}$, and $\boldsymbol{w}$) denote vectors. The $i$-th entry of $\boldsymbol{x}$ is noted by $\boldsymbol{x}^{(i)}$.
- $[N] := \{1, 2, \cdots, N\}$ denotes the index set from 1 to $N$.
- $[\boldsymbol{x}_N] = \{\boldsymbol{x} : \exists i \in [N] \ s.t. \ \boldsymbol{x} = \boldsymbol{x}_i\} \ (= \{\boldsymbol{x}_1, \boldsymbol{x}_2, \cdots, \boldsymbol{x}_N\} \ \text{if} \ (i \neq j \implies \boldsymbol{x}_i \neq \boldsymbol{x}_j))$.
- $\mathcal{H}$ is a task-specific Hilbert space, whose elements are called the *patch-like vectors*.
- Patches $\boldsymbol{x}$, features $\boldsymbol{v}$, and kernels $\boldsymbol{w}$ are all patch-like vectors.
- The images $\boldsymbol{z} = (\boldsymbol{x}_1, \boldsymbol{x}_2, \cdots, \boldsymbol{x}_P)^T \in \mathcal{H}^P$ are called $P$-patch data.
- $P$ is the number of patches in an image.
- $d$ is the dimension of $\mathcal{H}$, also known as the size of the patches/kernels.
- $C = c_2 d$ is the number of the kernels.
- $\mathcal{V} := \{\boldsymbol{v}_1, \boldsymbol{v}_2, \cdots, \boldsymbol{v}_d\}$ is the set of all features.
- $\mathcal{V}_{nor}$ and $\mathcal{V}_{aux}$ are the sets of normal and auxiliary features.
    - $\mathcal{V}_{nor} = \{\boldsymbol{v}_1, \boldsymbol{v}_2\}$ and $\mathcal{V}_{aux} = \{\boldsymbol{v}_3, \boldsymbol{v}_4, \cdots, \boldsymbol{v}_d\}$ in theoretical analysis.
    - We explore more combinations in the synthesized experiments.
- A patch $\boldsymbol{x}_i$ *contains* a feature $\boldsymbol{v}_k$ implies that $\boldsymbol{x}_i = \boldsymbol{v}_k + \rho$ for some small random noise (cf. Section 2.1). $\varepsilon$ controls the size of $\rho$.
- $i \in [P]$, $j \in [C]$, and $k \in [d]$ are the exclusive indices for patches ($\boldsymbol{x}_i$), kernels ($\boldsymbol{w}_j$), and features ($\boldsymbol{v}_k$), respectively.
- $\phi = \phi_d \circ \phi_e$ is the two-layer convolutional autoencoder defined in Section 2.2.
- $\sigma$ is the smoothed ReLU function defined in Equation (2), parameterized by $c_0$.
- $\ell(\boldsymbol{z}; \boldsymbol{w})$ and $\boldsymbol{R}_N(\boldsymbol{w})$ denote the reconstruction loss and error, respectively.
- $\sigma_0$ controls the size of the (zero-mean Gaussian) random initialization of $\boldsymbol{w}$.
- $T$ is the maximum number of iterations.
- Given $t \in [T]$ and $k \in [d]$, $\tilde{\mathcal{S}}^t(k)$ represents the cone set of feature $\boldsymbol{v}_k$ at iteration $t$.
- $f_r = f_r(d, t)$, $f_h = f_h(d)$, and constant $c_1$ controls the radius and height of the cone.
- $p_0 = p_0(d)$, $p_1 = p_1(d)$, and $p_2 = p_2(d)$ control the probabilistic bounds.
    - Equation (E.16) specify the expression of $p_0$. The upper bounds for $\|\boldsymbol{w}_j\|$ in Lemma 3.1 hold with probability at least $1 - p_0$.
    - $p_1$ (specified in Equation (E.28)) and $p_2$ (specified in Equation (E.29)) control the two-side bounds for $|\tilde{\mathcal{S}}^0(k)|$ (the initial size of the cone sets) in Theorem 2.

## D  PARAMETERS ANALYSIS

Selecting a set of proper parameters can significantly simplify the analyses of feature learning, especially those proofs that involve tight inequalities. However, finding such proper parameters is extremely laborious since every single attempt in a new set requires re-proving most of the results. Besides, we argue that fixing the parameters might confuse the readers and hurt the overall intuitiveness of the analysis. The choice of parameters is not unique in the analysis of feature learning; a wide range of different choices could lead to similar results. As a result, fixing the choice of parameters could conceal the actual relationship between different variables. For the above reasons, our paper specifies the constraints on the parameters instead of fixing them.

### D.1 RECOMMENDED PARAMETERS

The patch dimension $d$, number of patches $P$, and the probability $\beta_k$ ($k \in [d]$) in Assumption 2.1 are task-specific constants. It is convenient for the readers to take $P = 10$, $d = 20$, and $\beta_k < 0.5$ for all $k > 2$.

The other parameters in our paper can be divided into three groups:

1. The first group are constants, including $c_0$, $c_1$, and $c_2$. It is recommended to take $c_0 = 0.01$, $c_1 = 0.5$, and $c_2 = 1.2$.

2. The second group includes parameters that depend on the dimension $d$. Recall that we let $\|\boldsymbol{v}_k\| = 1$ for simplicity. To ensure that our analysis is without loss of generality, the noise threshold $\varepsilon$ and the variance of the random initialization $\sigma_0$ should be re-scaled accordingly. For better understanding, we suggest the readers consider $\sigma_0 \leq d^{-2}$ and $\varepsilon = \sigma_0$, e.g., $\varepsilon = \sigma = 0.001$, in this paper.

3. The third group includes parameters that depend on $t$ and $d$, mostly the variables related to the iterations. The constraints are discussed in the rest of this section.

We assume that the learning rate $\eta_t$ decays at the speed $\eta_t = 0.2t^{-1}$ as $t$ grow. For completeness, we let $\eta_0 = 0$ (i.e., the kernels stay unchanged in the first iteration). The proof of the results of our analysis imposes several constraints on the parameters. To ensure that Lemma 3.2 holds, we assume that

$$(c_1 - 2\sqrt{2\varepsilon})f_h(d) = (0.5 - 2\sqrt{0.002})f_h(d) > f_r(d, t), \tag{D.1}$$

which is a weak condition since $f_r(d, t) = o(t) \cdot f_h(d)$ and $\varepsilon \leq d^{-2}$. The proofs of Theorem 1 require $c_1, c_0$, and $\varepsilon$ to satisfy

$$c_1 - \sqrt{2\varepsilon} > c_0, \tag{D.2}$$

in which $c_0$ is the threshold in the smoothed ReLU function in Equation (2). Denote $\tilde{\varepsilon} = 3\sqrt{2\varepsilon} + 4\varepsilon \approx 0.138$, $\hat{\varepsilon} = 1 + \sqrt{2\varepsilon} \approx 1.045$, and $\tilde{c}_1 = c_1 - \sqrt{2\varepsilon} \approx 0.455$. As the main result of our analysis, Theorem 1 require

$$N\beta_k \left( 2\tilde{c}_1\hat{\varepsilon} - (1 + \sqrt{2\varepsilon})(\tilde{c}_1 + \hat{\varepsilon})(t\sigma_0 f_h)^2(c_2 - 1)d \right) \geq \frac{5c_1}{2}. \tag{D.3}$$

Taking the recommended parameters into Equation (D.3), we have

$$N\beta_k \left( 2 \times 0.455 \times 1.045 - 1.045 \times 1.5 \times 10^{-6} \times 4 \times (tf_h)^2 \right) \geq 1.25, \tag{D.4}$$

i.e., the first constraint on $N$, $\beta_k$, $t$, and $f_h$ is

$$N\beta_k \left( 0.951 - 6.27 \times 10^{-6}(tf_h)^2 \right) \geq 1.25. \tag{D.5}$$

Another constraint from Theorem 1 is

$$0.4N\beta_{k'}\sigma_0 \left( f_h \cdot (6\varepsilon + \sqrt{2\varepsilon} + \sqrt{2\varepsilon}t\sigma_0 f_h(c_2 - 1)d) + f_r \cdot \left( t\sigma_0 f_h \cdot (c_2 - 1)d + \sqrt{2\varepsilon} \right) \right)$$
$$\leq (t+1)\sigma_0 f_r(t+1, d). \tag{D.6}$$

Let $T = N = 100$, $f_h(d) = 0.2\sqrt{d}$, and $f_r(d, t) = f_h \cdot (t+1)^{-0.5}$. Equation (D.6) becomes

$$40\beta_{k'} \left( f_h \cdot (0.051 + 0.179 \times t\sigma_0 f_h) + f_r(4t\sigma_0 f_h + 0.045) \right) \leq (t+1)f_r(t+1, d) \tag{D.7}$$

It is not hard to check the validity of the recommended parameters.

## E  COMPLETE PROOFS

In this section, we go through the proofs in our analysis. We use a separate numbering for the equations in this section so that the readers do not need to turn back to the main parts to reference the equations. Denote

$$\Delta_{n,i} = \boldsymbol{x}_{n,i} - \phi_d^{(i)}(\phi_e(\boldsymbol{z}_n)), \tag{E.1}$$

i.e., the difference between the original patch $\boldsymbol{x}_{n,i}$ and the reconstructed patch $\phi_d^{(i)}(\phi_e(\boldsymbol{z}_n))$. Then, the empirical reconstruction error can be written as

$$\boldsymbol{R}_N(\boldsymbol{w}) = \frac{1}{N}\sum_{n\in[N]} \ell(\boldsymbol{z}_n; \boldsymbol{w}) = \frac{1}{N}\sum_{n\in[N]} \|\phi(\boldsymbol{z}_n; \boldsymbol{w}) - \boldsymbol{z}_n\|^2 = \frac{1}{N}\sum_{n\in[N]}\sum_{i\in[P]} \langle \Delta_{n,i}, \Delta_{n,i} \rangle. \tag{E.2}$$

As a warm-up, the following proposition computes the gradient of $\boldsymbol{R}_N(\boldsymbol{w})$.

*Proposition* 3.1. For all $j \in [C]$, the gradient of the empirical reconstruction error is

$$\nabla_{\boldsymbol{w}_j} \boldsymbol{R}_N(\boldsymbol{w}) = -\frac{2}{N} \sum_{n,i} \delta_n(i,j) \left[ \sigma(\langle \boldsymbol{w}_j, \boldsymbol{x}_{n,i} \rangle) \Delta_{n,i} + \langle \Delta_{n,i}, \boldsymbol{w}_j \rangle \, \sigma'(\langle \boldsymbol{w}_j, \boldsymbol{x}_{n,i} \rangle) \, \boldsymbol{x}_{n,i} \right]. \quad \text{(E.3)}$$

*Proof of Proposition* 3.1. By definition, we have

$$\nabla_{\boldsymbol{w}_j} \boldsymbol{R}_N(\boldsymbol{w}) = \nabla_{\boldsymbol{w}_j} \left( \frac{1}{N} \sum_{n \in N} \ell(\boldsymbol{z}_n) \right) = \frac{1}{N} \sum_{n \in [N]} \nabla_{\boldsymbol{w}_j} \ell(\boldsymbol{z}_n). \quad \text{(E.4)}$$

Recall

$$\phi(\boldsymbol{z}; \boldsymbol{w}) = \phi_d(\phi_e(\boldsymbol{z}; \boldsymbol{w})) = \left( \phi_d^{(1)}(\phi_e(\boldsymbol{z}; \boldsymbol{w})), \cdots, \phi_d^{(P)}(\phi_e(\boldsymbol{z}; \boldsymbol{w})) \right) \quad \text{(E.5)}$$

and

$$\ell(\boldsymbol{z}; \boldsymbol{w}) = \| \phi(\boldsymbol{z}; \boldsymbol{w}) - \boldsymbol{z} \|^2. \quad \text{(E.6)}$$

According to Equations (E.5) and (E.6), for all $n \in [N]$, we have

$$\ell(\boldsymbol{z}_n) = \| \phi(\boldsymbol{z}_n) - \boldsymbol{z}_n \|^2 = \sum_{i \in P} \left\langle \phi_d^{(i)}(\phi_e(\boldsymbol{z}_n)) - \boldsymbol{x}_{n,i}, \phi_d^{(i)}(\phi_e(\boldsymbol{z}_n)) - \boldsymbol{x}_{n,i} \right\rangle = \sum_{i \in [P]} \langle \Delta_{n,i}, \Delta_{n,i} \rangle, \quad \text{(E.7)}$$

and the gradient of $\ell(\boldsymbol{z}_n)$ can be written as

$$\nabla_{\boldsymbol{w}_j} \left( \sum_{i \in P} \langle \Delta_{n,i}, \Delta_{n,i} \rangle \right) = \sum_{i \in P} 2 \langle \nabla_{\boldsymbol{w}_j} \Delta_{n,i}, \Delta_{n,i} \rangle \quad \text{(E.8)}$$

Here, we abuse the notion of inner product $\langle \nabla_{\boldsymbol{w}_j} \Delta_{n,i}, \Delta_{n,i} \rangle$ to simplify the calculation of the gradient of $\ell$ without causing confusion. Consider the right-hand side (RHS) of Equation (E.8). For all $i \in [P]$, the term in the sum equal to

$$\langle \nabla_{\boldsymbol{w}_j} \Delta_{n,i}, \Delta_{n,i} \rangle = \left\langle \Delta_{n,i}, \nabla_{\boldsymbol{w}_j} \left( \boldsymbol{x}_{n,i} - \sum_{j' \in C} \sigma(\langle \boldsymbol{w}_{j'}, \boldsymbol{x}_{n,i} \rangle) \delta_n(i,j') \boldsymbol{w}_{j'} \right) \right\rangle$$

$$= -\left\langle \Delta_{n,i}, \nabla_{\boldsymbol{w}_j} \left( \sum_{j' \in \mathcal{S}_n(i)} \sigma(\langle \boldsymbol{w}_{j'}, \boldsymbol{x}_{n,i} \rangle) \boldsymbol{w}_{j'} \right) \right\rangle. \quad \text{(E.9)}$$

Here, $\delta_n(i,j)$ is the natural extension of $\delta(i,j)$, considering that $\boldsymbol{x}_i$ is replaced by $\boldsymbol{x}_{n,i}$ in the proof. $\mathcal{S}_n(i)$ is defined as

$$\mathcal{S}_n(i) := \{ j \in [C] : \delta_n(i,j) = 1 \}. \quad \text{(E.10)}$$

$\mathcal{S}_n(i)$ can be interpreted as the "empirical cone set of the patch $\boldsymbol{x}_i$". From Equation (E.9) we know that $\langle \nabla_{\boldsymbol{w}_j} \Delta_{n,i}, \Delta_{n,i} \rangle$ takes 0 when $j \notin \mathcal{S}_n(i)$. Otherwise, we have

$$-\langle \nabla_{\boldsymbol{w}_j} \Delta_{n,i}, \Delta_{n,i} \rangle = \langle \Delta_{n,i}, \nabla_{\boldsymbol{w}_j} (\sigma(\langle \boldsymbol{w}_j, \boldsymbol{x}_{n,i} \rangle) \boldsymbol{w}_j) \rangle$$

$$= \sigma(\langle \boldsymbol{w}_j, \boldsymbol{x}_{n,i} \rangle) \Delta_{n,i} + \langle \Delta_{n,i}, \boldsymbol{w}_j \rangle \sigma'(\langle \boldsymbol{w}_j, \boldsymbol{x}_{n,i} \rangle) \boldsymbol{x}_{n,i}, \quad \text{(E.11)}$$

in which for all $u \in \mathbb{R}$, the derivative of $\sigma$ at $u$ is

$$\sigma'(u) = \begin{cases} 0 & \text{if } u \leq 0; \\ (c_0)^{-1} x & \text{if } 0 < u \leq c_0; \\ 1 & \text{if } u > c_0. \end{cases} \quad \text{(E.12)}$$

Finally, we can obtain Equation (E.3) by combining Equations (E.4) to (E.12), which completes the proof. □

After random initialization, the kernels iterate according to the gradient descent rule, i.e.,

$$\boldsymbol{w}_j^{t+1} = \boldsymbol{w}_j^t - \eta_t \nabla_{\boldsymbol{w}_j} \boldsymbol{R}_N(\boldsymbol{w}^t). \quad \text{(E.13)}$$

Let $f_h = f_h(d)$ be the control function of the growth of the major part of the kernels (i.e., the height of the cone in our cone set intuition). At each iteration step, the size of the kernels can be bounded by the following lemma.

*Lemma* 3.1. With probability at least $1 - p_0$, the random initialization of $\boldsymbol{w}_j^0$ satisfies

$$\|\boldsymbol{w}_j^0\| \le \sigma_0 f_h. \tag{E.14}$$

For all $t \ge 1$, the size of $\boldsymbol{w}_j^t$ is controlled by

$$\|\boldsymbol{w}_j^t\| \le t\sigma_0 f_h. \tag{E.15}$$

with probability at least $1 - p_0$. In both Equations (E.14) and (E.15), we let $p_0$ take

$$p_0 = 2\exp\left(-\frac{(f_h - d)^2}{8d}\right). \tag{E.16}$$

*Proof of Lemma* 3.1. By definition, we can write

$$\boldsymbol{w}_j^0 = \left(\boldsymbol{w}_j^{0,(1)}, \boldsymbol{w}_j^{0,(2)}, \cdots, \boldsymbol{w}_j^{0,(d)}\right)^T \sim \mathcal{N}(0, \sigma_0 \boldsymbol{I}) \tag{E.17}$$

and

$$\frac{1}{\sigma_0^2}\|\boldsymbol{w}_j^0\|^2 = \sum_{i \in [d]} \left(\frac{\boldsymbol{w}_j^{0,(i)}}{\sigma_0}\right)^2 \sim \chi_d^2, \tag{E.18}$$

in which $\chi_d^2$ is the chi-squared distribution with $d$ degrees of freedom. Lemma F.3 implies that

$$\boldsymbol{P}\left[\left|\|\boldsymbol{w}_j^0\|^2 - \sigma_0^2 d\right| \ge \lambda\sigma_0^2 d\right] \le 2e^{-\lambda^2 d/8} \tag{E.19}$$

for all $\lambda \in (0, 1)$. We can obtain Equation (E.14) by letting

$$\lambda = \frac{f_h^2 - d}{d}. \tag{E.20}$$

We prove the second result by induction. Equation (E.19) implies that $\|\boldsymbol{w}_j^1\| = \|\boldsymbol{w}_j^0\| \le \sigma_0 f_h$ as required. For all $t > 1$, assume that $\boldsymbol{w}_j^{t-1}$ satisfies Equation (E.15). By definition, we have

$$
\begin{aligned}
&\|\boldsymbol{w}_j^t\| \\
=& \left\|\boldsymbol{w}_j^{t-1} - \eta_{t-1}\nabla_{\boldsymbol{w}_j^{t-1}}\boldsymbol{R}_N(\boldsymbol{w}^{t-1})\right\| \\
=& \left\|\boldsymbol{w}_j + \frac{\eta}{N}\sum_{n \in [N]}\sum_{i \in [P]} 2\delta_n(i,j)\left[\sigma\left(\langle\boldsymbol{w}_j, \boldsymbol{x}_{n,i}\rangle\right)\Delta_{n,i} + \langle\Delta_{n,i}, \boldsymbol{w}_j\rangle\sigma'\left(\langle\boldsymbol{w}_j, \boldsymbol{x}_{n,i}\rangle\right)\boldsymbol{x}_{n,i}\right]\right\| \\
\le& \|\boldsymbol{w}_j\| + \left\|\frac{\eta}{N}\sum_{n \in [N]}\sum_{i \in [P]} 2\delta_n(i,j)\left[\sigma\left(\langle\boldsymbol{w}_j, \boldsymbol{x}_{n,i}\rangle\right)\Delta_{n,i} + \langle\Delta_{n,i}, \boldsymbol{w}_j\rangle\sigma'\left(\langle\boldsymbol{w}_j, \boldsymbol{x}_{n,i}\rangle\right)\boldsymbol{x}_{n,i}\right]\right\|.
\end{aligned} \tag{E.21}
$$

Due to limited space, the subscript or superscript of $t - 1$ is omitted without causing any confusion. For all $n \in [N]$ and $\forall i \in [P]$, it is easy to obtain that

$$
\begin{aligned}
&\|\sigma\left(\langle\boldsymbol{w}_j, \boldsymbol{x}_{n,i}\rangle\right)\Delta_{n,i} + \langle\Delta_{n,i}, \boldsymbol{w}_j\rangle\sigma'\left(\langle\boldsymbol{w}_j, \boldsymbol{x}_{n,i}\rangle\right)\boldsymbol{x}_{n,i}\| \\
\le& 2\|\boldsymbol{x}_{n,i}\|\|\Delta_{n,i}\|\|\boldsymbol{w}_j\| \\
\le& 2(1 + \varepsilon)^2\|\boldsymbol{w}_j\|.
\end{aligned} \tag{E.22}
$$

Since only one of the $i \in [P]$ is counted in the second summation in Equation (E.21) for all $n \in [N]$ due to the property of $\delta_n(i,j)$, we have

$$\left\|\frac{\eta}{N}\sum_{n \in [N]}\sum_{i \in [P]} 2\delta_n(i,j)[\sigma(\langle\boldsymbol{w}_j, \boldsymbol{x}_{n,i}\rangle)\Delta_{n,i} + \langle\Delta_{n,i}, \boldsymbol{w}_j\rangle\sigma'(\langle\boldsymbol{w}_j, \boldsymbol{x}_{n,i}\rangle)\boldsymbol{x}_{n,i}]\right\| \le 4(1 + \varepsilon)^2\eta\|\boldsymbol{w}_j\|,$$

which implies that

$$\|\boldsymbol{w}_j^t\| \le (1 + 5\eta_{t-1})\|\boldsymbol{w}_j^{t-1}\|. \tag{E.23}$$

We can obtain Equation (E.15) by taking $\eta_t = 0.2t^{-1}$ into Equation (E.23), which finishes the proof of this lemma. □

*Lemma* 3.2. For all $t \geq 1$, if $j \in \tilde{\mathcal{S}}^t(k)$ and $\boldsymbol{x}_i$ (or $\boldsymbol{x}_{n,i}$) contains $\boldsymbol{v}_k$, then we have $\delta(i, j) = 1$ (or $\delta_n(i, j) = 1$).

*Proof.* Recall that $\delta(i, j)$ is a binary-value function that takes 1 when

$$i = \arg\max_{i' \in [P]} \sigma(\langle \boldsymbol{w}_j^t, \boldsymbol{x}_{i'} \rangle) \tag{E.24}$$

and 0 otherwise. Since $\sigma$ is non-decreasing, we have

$$\arg\max_{i \in [P]} \sigma(\langle \boldsymbol{w}_j, \boldsymbol{x}_i \rangle) = \arg\max_{i \in [P]} \langle \boldsymbol{w}_j, \boldsymbol{x}_i \rangle. \tag{E.25}$$

When $j \in \tilde{\mathcal{S}}^t(k)$ and $\boldsymbol{x}_i$ contains $\boldsymbol{v}_k$, $\langle \boldsymbol{w}_j, \boldsymbol{x}_i \rangle$ can be bounded by

$$\begin{aligned} \langle \boldsymbol{w}_j^t, \boldsymbol{x}_i \rangle &= \langle \boldsymbol{w}_j^t, \boldsymbol{v}_k \rangle + \langle \boldsymbol{w}_j^t, \boldsymbol{x}_i - \boldsymbol{v}_k \rangle \\ &\geq t\sigma_0 c_1 f_h - \|\boldsymbol{w}_j^t\| \|\boldsymbol{x}_i - \boldsymbol{v}_k\| \\ &\geq (c_1 - \sqrt{2\varepsilon}) t\sigma_0 f_h. \end{aligned} \tag{E.26}$$

On the other hand, consider $i'$ and $k'$ such that $\boldsymbol{x}_{i'}$ contains $\boldsymbol{v}_{k'}$. By Assumption 2.1, $k' \neq k$ if given $i' \neq i$. Therefore, we have

$$\begin{aligned} \langle \boldsymbol{w}_j, \boldsymbol{x}_{i'} \rangle &= \langle \boldsymbol{w}_j, \boldsymbol{v}_{k'} \rangle + \langle \boldsymbol{w}_j, \boldsymbol{x}_{i'} - \boldsymbol{v}_{k'} \rangle \\ &\leq t\sigma_0 f_r + \|\boldsymbol{w}_j\| \|\boldsymbol{x}_i - \boldsymbol{v}_k\| \\ &\leq (f_r + \sqrt{2\varepsilon} f_h) t\sigma_0. \end{aligned} \tag{E.27}$$

We prove the lemma by combining Equations (E.26) and (E.27). $\qquad\square$

Denote

$$p_1 = \left( \frac{c_1 f_h}{(c_1 f_h)^2 + 1} \frac{e^{-(c_1 f_h)^2}}{\sqrt{2\pi}} \right) \left( 1 - \frac{1}{f_r} \frac{e^{-f_r^2}}{\sqrt{2\pi}} \right)^{C-1} \tag{E.28}$$

and

$$p_2 = \left( \frac{1}{c_1 f_h} \frac{e^{-(c_1 f_h)^2}}{\sqrt{2\pi}} \right) \left( 1 - \frac{f_r}{f_r^2 + 1} \frac{e^{-f_r^2}}{\sqrt{2\pi}} \right)^{C-1}. \tag{E.29}$$

The following lemma proves that the size of $\tilde{\mathcal{S}}$ at random initialization is bounded from both sides. Note that $\boldsymbol{w}^0 = \boldsymbol{w}^1$ since we set $\eta_0 = 0$.

*Lemma* 2. Let $p_1$ and $p_2$ be parameters depending on $d$ and $t$. For all $k \in [d]$, the size of $\tilde{\mathcal{S}}^0(k)$ can be upper-bounded by

$$|\tilde{\mathcal{S}}^0(k)| \leq Cp_2 + \lambda, \tag{E.30}$$

with probability at least $1 - \exp\left(-\frac{3\lambda^2}{6Cp_2 + 2\lambda}\right)$ over random initialization. The lower-bound

$$|\tilde{\mathcal{S}}^0(k)| \geq Cp_1 - \lambda \tag{E.31}$$

hold with probability at least $1 - \exp\left(-\frac{\lambda^2}{2Cp_2}\right)$.

*Proof of Theorem* 2. According to Dümbgen (2010), we have

$$\frac{1}{x + 1/x} \frac{e^{-x^2}}{\sqrt{2\pi}} < \mathop{\boldsymbol{P}}_{g \sim \mathcal{N}(0,1)} [g > x] < \frac{1}{x} \frac{e^{-x^2}}{\sqrt{2\pi}} \tag{E.32}$$

for arbitrary $x > 0$. For all $j \in [C]$, we have

$$\frac{\langle \boldsymbol{w}_j^1, \boldsymbol{v}_k \rangle}{\sigma_0} = \sum_{k \in [d]} \boldsymbol{v}_k^{(k)} \frac{\boldsymbol{w}_j^{1,(k)}}{\sigma_0} \sim \mathcal{N}(0, 1). \tag{E.33}$$

Take $x = c_1 f_h$ in Equation (E.32), we can obtain

$$\frac{c_1 f_h}{(c_1 f_h)^2 + 1} \frac{e^{-(c_1 f_h)^2}}{\sqrt{2\pi}} < \boldsymbol{P}_{\boldsymbol{w}_j^0} \left[ \langle \boldsymbol{w}_j^1, \boldsymbol{v}_k \rangle > \sigma_0 c_1 f_h \right] < \frac{1}{c_1 f_h} \frac{e^{-(c_1 f_h)^2}}{\sqrt{2\pi}}. \quad \text{(E.34)}$$

Similarly, we have

$$\frac{f_r}{f_r^2 + 1} \frac{e^{-f_r^2}}{\sqrt{2\pi}} < \boldsymbol{P}_{\boldsymbol{w}_j^0} \left[ \langle \boldsymbol{w}_j^1, \boldsymbol{v}_k \rangle > \sigma_0 f_r \right] < \frac{1}{f_r} \frac{e^{-f_r^2}}{\sqrt{2\pi}}. \quad \text{(E.35)}$$

by letting $x = f_r$. By definition, we have

$$\boldsymbol{P}_{\boldsymbol{w}^0} \left[ j \in \tilde{\mathcal{S}}^0(k) \right] = \boldsymbol{P}_{\boldsymbol{w}_j^0} \left[ \langle \boldsymbol{w}_j^1, \boldsymbol{v}_k \rangle > \sigma_0 c_1 f_h \right] \cdot \left( 1 - \boldsymbol{P}_{\boldsymbol{w}_j^0} \left[ \langle \boldsymbol{w}_j^1, \boldsymbol{v}_k \rangle > \sigma_0 f_r \right] \right)^{C-1}. \quad \text{(E.36)}$$

Combining Equations (E.34) to (E.36), we obtain that

$$p_1 \leq \boldsymbol{P}_{\boldsymbol{w}_j^0} \left[ j \in \tilde{\mathcal{S}}^0(k) \right] \leq p_2 \quad \text{(E.37)}$$

By Corollary F.1 we immediately get Equations (13) and (14). $\qquad\square$

*Theorem* 1. Assume that $|\tilde{\mathcal{S}}^0(k)| \geq 1$ for all $k \in [d]$. For all $t \in [T]$ and $\forall k \in [d]$, we have $\tilde{\mathcal{S}}^t(k) \subseteq \tilde{\mathcal{S}}^{t+1}(k)$.

*Proof of Theorem 1.* We prove $\tilde{\mathcal{S}}^t(k) \subseteq \tilde{\mathcal{S}}^{t+1}(k)$ by induction. For any $j \in [C]$, assume that $\boldsymbol{w}_j \in \tilde{\mathcal{S}}^t(k)$ at iteration $t$. The growth of $\boldsymbol{w}_j^t$ along $\boldsymbol{v}_k$ is given by

$$\begin{aligned}
\langle \boldsymbol{w}_j^{t+1} - \boldsymbol{w}_j^t, \boldsymbol{v}_k \rangle &= \langle -\eta_t \nabla_{\boldsymbol{w}_j^t} \boldsymbol{R}_N(\boldsymbol{w}), \boldsymbol{v}_k \rangle \\
&= \sum_{n \in [N]} \sum_{i \in [P]} 2\eta_t \delta_n(i,j) \left[ \sigma\left( \langle \boldsymbol{w}_j^t, \boldsymbol{x}_{n,i} \rangle \right) \langle \Delta_{n,i}, \boldsymbol{v}_k \rangle + \langle \Delta_{n,i}, \boldsymbol{w}_j^t \rangle \sigma'\left( \langle \boldsymbol{w}_j^t, \boldsymbol{x}_{n,i} \rangle \right) \langle \boldsymbol{x}_{n,i}, \boldsymbol{v}_k \rangle \right]
\end{aligned}$$
$$\text{(E.38)}$$

Given $n \in [N]$, suppose $\exists \boldsymbol{x}_{n,i_n}$ such that $\boldsymbol{x}_{n,i_n}$ contains $\boldsymbol{v}_k$. By definition, this condition holds for $\beta_k N$ many training samples. Then, Lemma 3.2 implies that

$$\begin{aligned}
&\sum_{i \in [P]} \delta_n(i,j) \left[ \sigma\left( \langle \boldsymbol{w}_j^t, \boldsymbol{x}_{n,i} \rangle \right) \langle \Delta_{n,i}, \boldsymbol{v}_k \rangle + \langle \Delta_{n,i}, \boldsymbol{w}_j^t \rangle \sigma'\left( \langle \boldsymbol{w}_j^t, \boldsymbol{x}_{n,i} \rangle \right) \langle \boldsymbol{x}_{n,i}, \boldsymbol{v}_k \rangle \right] \\
&= \sigma\left( \langle \boldsymbol{w}_j^t, \boldsymbol{x}_{n,i_n} \rangle \right) \langle \Delta_{n,i_n}, \boldsymbol{v}_k \rangle + \langle \Delta_{n,i_n}, \boldsymbol{w}_j^t \rangle \sigma'\left( \langle \boldsymbol{w}_j^t, \boldsymbol{x}_{n,i_n} \rangle \right) \langle \boldsymbol{x}_{n,i_n}, \boldsymbol{v}_k \rangle .
\end{aligned}$$
$$\text{(E.39)}$$

Otherwise, given $n \in [N]$, we let $\boldsymbol{x}_{n,i_n} := 0$ when there is no $\boldsymbol{x}_{n,i}$ such that $\boldsymbol{x}_{n,i}$ contains $\boldsymbol{v}_k$ with slight abuse of notation. Using these notations, we can write

$$\langle \boldsymbol{w}_j^{t+1} - \boldsymbol{w}_j^t, \boldsymbol{v}_k \rangle = 2\eta_t \sum_{n \in [N]} \sigma\left( \langle \boldsymbol{w}_j^t, \boldsymbol{x}_{n,i_n} \rangle \right) \langle \Delta_{n,i_n}, \boldsymbol{v}_k \rangle + \langle \Delta_{n,i_n}, \boldsymbol{w}_j^t \rangle \sigma'\left( \langle \boldsymbol{w}_j^t, \boldsymbol{x}_{n,i_n} \rangle \right) \langle \boldsymbol{x}_{n,i_n}, \boldsymbol{v}_k \rangle$$
$$\text{(E.40)}$$

It remains to bound the terms in Equation (E.40). We first look at $\langle \Delta_{n,i_n}, \boldsymbol{v}_k \rangle$.

$$\begin{aligned}
\langle \Delta_{n,i_n}, \boldsymbol{v}_k \rangle &= \left\langle \left( \boldsymbol{x}_{n,i_n} - \sum_{j' \in \mathcal{S}_n(i_n)} \sigma\left( \langle \boldsymbol{w}_{j'}^t, \boldsymbol{x}_{n,i_n} \rangle \right) \boldsymbol{w}_{j'}^t \right), \boldsymbol{v}_k \right\rangle \\
&= \langle \boldsymbol{x}_{n,i_n}, \boldsymbol{v}_k \rangle - \left\langle \sum_{j' \in \mathcal{S}_n(i_n)} \sigma\left( \langle \boldsymbol{w}_{j'}^t, \boldsymbol{x}_{n,i_n} \rangle \right) \boldsymbol{w}_{j'}^t, \boldsymbol{v}_k \right\rangle \\
&= \langle \boldsymbol{x}_{n,i_n}, \boldsymbol{v}_k \rangle - \sum_{j' \in \mathcal{S}_n(i_n)} \langle \sigma\left( \langle \boldsymbol{w}_{j'}^t, \boldsymbol{x}_{n,i_n} \rangle \right) \boldsymbol{w}_{j'}^t, \boldsymbol{v}_k \rangle
\end{aligned}$$
$$\text{(E.41)}$$

Apparently, both sides of Equation (E.41) would take zero if $\boldsymbol{x}_{n,i_n} = 0$. We only need to consider the case when $\boldsymbol{x}_{n,i_n} \neq 0$. Since $\boldsymbol{x}_{n,i_n}$ contains $\boldsymbol{v}_k$, for all $j' \in \mathcal{S}_n(i_n)$, we have

$$
\begin{aligned}
\sigma\left(\left\langle \boldsymbol{w}_{j'}^t, \boldsymbol{x}_{n,i_n} \right\rangle\right) &\leq \left\langle \boldsymbol{w}_{j'}^t, \boldsymbol{x}_{n,i_n} \right\rangle \\
&= \left\langle \boldsymbol{w}_{j'}^t, \boldsymbol{v}_k \right\rangle + \left\langle \boldsymbol{w}_{j'}^t, \boldsymbol{x}_{n,i_n} - \boldsymbol{v}_k \right\rangle \\
&\leq \left\langle \boldsymbol{w}_{j'}^t, \boldsymbol{v}_k \right\rangle + \sqrt{2\varepsilon}\|\boldsymbol{w}_{j'}^t\| \\
&\leq (1 + \sqrt{2\varepsilon})t\sigma_0 f_h.
\end{aligned}
\tag{E.42}
$$

Lemma 3.2 implies that Equation (E.42) also hold for $\sigma\left(\left\langle \boldsymbol{w}_j^t, \boldsymbol{x}_{n,i_n} \right\rangle\right)$ since $j \in \tilde{\mathcal{S}}^t(k)$. Combining Equations (E.41) and (E.42), we have

$$
\left\langle \Delta_{n,i}, \boldsymbol{v}_k \right\rangle \geq 1 - \varepsilon - (1 + \sqrt{2\varepsilon}) \sum_{j' \in \mathcal{S}_n(i_n)} \|\boldsymbol{w}_{j'}^t\|^2 \geq 1 - \varepsilon - (1 + \sqrt{2\varepsilon})(t\sigma_0 f_h)^2 |\mathcal{S}_n^t(i_n)|. \tag{E.43}
$$

Since we assume $\tilde{\mathcal{S}}^0(k) \geq 1$ for all features, we have

$$
|\mathcal{S}_n^t(i_n)| \leq (c_2 - 1)d. \tag{E.44}
$$

Next, we turn to $\left\langle \Delta_{n,i}, \boldsymbol{w}_j^t \right\rangle$. Similarly, we have

$$
\begin{aligned}
\left\langle \Delta_{n,i}, \boldsymbol{w}_j^t \right\rangle &= \left\langle \left( \boldsymbol{x}_{n,i} - \sum_{j' \in \mathcal{S}_{n,i}} \sigma\left(\left\langle \boldsymbol{w}_{j'}^t, \boldsymbol{x}_{n,i} \right\rangle\right) \boldsymbol{w}_{j'}^t \right), \boldsymbol{w}_j^t \right\rangle \\
&= \left\langle \boldsymbol{x}_{n,i}, \boldsymbol{w}_j^t \right\rangle - \left\langle \sum_{j' \in \mathcal{S}_{n,i}} \sigma\left(\left\langle \boldsymbol{w}_{j'}^t, \boldsymbol{x}_{n,i} \right\rangle\right) \boldsymbol{w}_{j'}^t, \boldsymbol{w}_j^t \right\rangle \\
&= \left\langle \boldsymbol{x}_{n,i}, \boldsymbol{w}_j^t \right\rangle - \sum_{j' \in \mathcal{S}_{n,i}} \left\langle \sigma\left(\left\langle \boldsymbol{w}_{j'}^t, \boldsymbol{x}_{n,i} \right\rangle\right) \boldsymbol{w}_{j'}^t, \boldsymbol{w}_j^t \right\rangle \\
&\geq \left\langle \boldsymbol{w}_j^t, \boldsymbol{v}_k \right\rangle - \sqrt{2\varepsilon}\|\boldsymbol{w}_j^t\| - (1 + \sqrt{2\varepsilon}) \sum_{j' \in \mathcal{S}_{n,i}} \|\boldsymbol{w}_{j'}^t\|^2 \|\boldsymbol{w}_j^t\| \\
&\geq t\sigma_0 f_h \cdot \left( c_1 - \sqrt{2\varepsilon} - (1 + \sqrt{2\varepsilon})(t\sigma_0 f_h)^2 |\mathcal{S}_n^t(i_n)| \right).
\end{aligned}
\tag{E.45}
$$

Finally, we derive the lower bound of $\left\langle \boldsymbol{w}_j^t, \boldsymbol{x}_{n,i_n} \right\rangle$ by

$$
\begin{aligned}
\left\langle \boldsymbol{w}_j^t, \boldsymbol{x}_{n,i_n} \right\rangle &= \left\langle \boldsymbol{w}_j^t, \boldsymbol{v}_k \right\rangle + \left\langle \boldsymbol{w}_j^t, \boldsymbol{x}_{n,i_n} - \boldsymbol{v}_k \right\rangle \\
&\geq \left\langle \boldsymbol{w}_j^t, \boldsymbol{v}_k \right\rangle - \sqrt{2\varepsilon}\|\boldsymbol{w}_j^t\| \\
&\geq (c_1 - \sqrt{2\varepsilon})t\sigma_0 f_h,
\end{aligned}
\tag{E.46}
$$

which also implies that $\sigma\left(\left\langle \boldsymbol{w}_j^t, \boldsymbol{x}_{n,i_n} \right\rangle\right) \geq (c_1 - \sqrt{2\varepsilon})t\sigma_0 f_h$ and $\sigma'\left(\left\langle \boldsymbol{w}_j^t, \boldsymbol{x}_{n,i_n} \right\rangle\right) = 1$.

Combining all above, we have

$$
\begin{aligned}
\frac{1}{2\eta_t} &\left\langle \boldsymbol{w}_j^{t+1} - \boldsymbol{w}_j^t, \boldsymbol{v}_k \right\rangle \\
&= \sum_{n \in [N]} \sigma\left(\left\langle \boldsymbol{w}_j^t, \boldsymbol{x}_{n,i_n} \right\rangle\right) \left\langle \Delta_{n,i_n}, \boldsymbol{v}_k \right\rangle + \left\langle \Delta_{n,i_n}, \boldsymbol{w}_j^t \right\rangle \sigma'\left(\left\langle \boldsymbol{w}_j^t, \boldsymbol{x}_{n,i_n} \right\rangle\right) \left\langle \boldsymbol{x}_{n,i_n}, \boldsymbol{v}_k \right\rangle \\
&\geq N\beta_k \left( (c_1 - \sqrt{2\varepsilon})t\sigma_0 f_h \left( 1 - \varepsilon - (1 + \sqrt{2\varepsilon})(t\sigma_0 f_h)^2 |\mathcal{S}_n^t(i_n)| \right) \right) \\
&\quad + (1 - \varepsilon)N\beta_k t\sigma_0 f_h \cdot \left( c_1 - \sqrt{2\varepsilon} - (1 + \sqrt{2\varepsilon})(t\sigma_0 f_h)^2 |\mathcal{S}_n^t(i_n)| \right),
\end{aligned}
\tag{E.47}
$$

which implies that

$$
\left\langle \boldsymbol{w}_j^{t+1}, \boldsymbol{v}_k \right\rangle \geq c_1 \sigma_0 f_h \left( t + 2c_1^{-1}t\eta_t N\beta_k \left( 2\tilde{c}_1\hat{\varepsilon} - (1 + \sqrt{2\varepsilon})(\tilde{c}_1 + \hat{\varepsilon})(t\sigma_0 f_h)^2 \cdot (c_2 - 1)d \right) \right)
\tag{E.48}
$$

As discussed in Appendix D,

$$2c_1^{-1}t\eta_t N\beta_k \left(2\tilde{c}_1\hat{\varepsilon} - (1+\sqrt{2\varepsilon})(\tilde{c}_1+\hat{\varepsilon})(t\sigma_0 f_h)^2 \cdot (c_2-1)d\right) \leq 1 \tag{E.49}$$

is not a tight bound. Under mild conditions on the choice of the parameters, we can obtain

$$\langle \boldsymbol{w}_j^{t+1}, \boldsymbol{v}_k \rangle \geq (t+1)c_1\sigma_0 f_h \tag{E.50}$$

as expected.

Next, for all $k' \neq k$, from Definition 3.1 we know that

$$\langle \boldsymbol{w}_j^t, \boldsymbol{v}_{k'} \rangle \leq t\sigma_0 f_r. \tag{E.51}$$

For all $j \in [C]$, the growth of $\boldsymbol{w}_j^t$ along $\boldsymbol{v}_{k'}$ is given by

$$\begin{aligned}
&\langle \boldsymbol{w}_j^{t+1} - \boldsymbol{w}_j^t, \boldsymbol{v}_{k'} \rangle \\
&= \sum_{n\in[N]}\sum_{i\in[P]} 2\eta_t \delta_n(i,j)\left[\sigma\left(\langle \boldsymbol{w}_j^t, \boldsymbol{x}_{n,i}\rangle\right)\langle \Delta_{n,i}, \boldsymbol{v}_{k'}\rangle + \langle \Delta_{n,i}, \boldsymbol{w}_j^t\rangle\sigma'\left(\langle \boldsymbol{w}_j^t, \boldsymbol{x}_{n,i}\rangle\right)\langle \boldsymbol{x}_{n,i}, \boldsymbol{v}_{k'}\rangle\right]
\end{aligned} \tag{E.52}$$

Note that the evaluation of $\delta_n(i,j)$ is independent of $k$ (or $k'$). Therefore, similar to Equations (E.39) and (E.40), we have

$$\begin{aligned}
&\sum_{i\in[P]} \delta_n(i,j)\left[\sigma\left(\langle \boldsymbol{w}_j^t, \boldsymbol{x}_{n,i}\rangle\right)\langle \Delta_{n,i}, \boldsymbol{v}_{k'}\rangle + \langle \Delta_{n,i}, \boldsymbol{w}_j^t\rangle\sigma'\left(\langle \boldsymbol{w}_j^t, \boldsymbol{x}_{n,i}\rangle\right)\langle \boldsymbol{x}_{n,i}, \boldsymbol{v}_{k'}\rangle\right] \\
&= \sigma\left(\langle \boldsymbol{w}_j^t, \boldsymbol{x}_{n,i_n}\rangle\right)\langle \Delta_{n,i_n}, \boldsymbol{v}_{k'}\rangle + \langle \Delta_{n,i_n}, \boldsymbol{w}_j^t\rangle\sigma'\left(\langle \boldsymbol{w}_j^t, \boldsymbol{x}_{n,i_n}\rangle\right)\langle \boldsymbol{x}_{n,i_n}, \boldsymbol{v}_{k'}\rangle
\end{aligned} \tag{E.53}$$

and

$$\langle \boldsymbol{w}_j^{t+1} - \boldsymbol{w}_j^t, \boldsymbol{v}_{k'} \rangle = 2\eta_t \sum_{n\in[N]} \sigma\left(\langle \boldsymbol{w}_j^t, \boldsymbol{x}_{n,i_n}\rangle\right)\langle \Delta_{n,i_n}, \boldsymbol{v}_{k'}\rangle + \langle \Delta_{n,i_n}, \boldsymbol{w}_j^t\rangle\sigma'\left(\langle \boldsymbol{w}_j^t, \boldsymbol{x}_{n,i_n}\rangle\right)\langle \boldsymbol{x}_{n,i_n}, \boldsymbol{v}_{k'}\rangle. \tag{E.54}$$

In Equations (E.53) and (E.54), $\boldsymbol{x}_{n,i_n}$ refers to the patch that contain $\boldsymbol{v}_{k'}$ if $\exists i \in [P]$ such that $\boldsymbol{x}_{n,i}$ contains $\boldsymbol{v}_{k'}$ or $\boldsymbol{x}_{n,i_n} = 0$ otherwise (cf. Equation (E.39)). It remains to upper-bound the terms in Equation (E.54)

$$\begin{aligned}
\langle \Delta_{n,i_n}, \boldsymbol{v}_{k'} \rangle &= \left\langle \left(\boldsymbol{x}_{n,i_n} - \sum_{j'\in\mathcal{S}_n(i_n)} \sigma\left(\langle \boldsymbol{w}_{j'}^t, \boldsymbol{x}_{n,i_n}\rangle\right)\boldsymbol{w}_{j'}^t\right), \boldsymbol{v}_{k'} \right\rangle \\
&= \langle \boldsymbol{x}_{n,i_n}, \boldsymbol{v}_{k'} \rangle - \left\langle \sum_{j'\in\mathcal{S}_n(i_n)} \sigma\left(\langle \boldsymbol{w}_{j'}^t, \boldsymbol{x}_{n,i_n}\rangle\right)\boldsymbol{w}_{j'}^t, \boldsymbol{v}_{k'} \right\rangle \\
&= \langle \boldsymbol{x}_{n,i_n}, \boldsymbol{v}_{k'} \rangle - \sum_{j'\in\mathcal{S}_n(i_n)} \left\langle \sigma\left(\langle \boldsymbol{w}_{j'}^t, \boldsymbol{x}_{n,i_n}\rangle\right)\boldsymbol{w}_{j'}^t, \boldsymbol{v}_{k'} \right\rangle,
\end{aligned} \tag{E.55}$$

in which

$$\langle \boldsymbol{x}_{n,i_n}, \boldsymbol{v}_{k'} \rangle = \langle \boldsymbol{x}_{n,i_n} - \boldsymbol{v}_k, \boldsymbol{v}_{k'} \rangle + \langle \boldsymbol{v}_k, \boldsymbol{v}_{k'} \rangle = \langle \boldsymbol{x}_{n,i_n} - \boldsymbol{v}_k, \boldsymbol{v}_{k'} \rangle \leq \sqrt{2\varepsilon}. \tag{E.56}$$

By definition, we have $\langle \boldsymbol{w}_{j'}^t, \boldsymbol{v}_{k'} \rangle \geq 0$ for all $j' \in \mathcal{S}_n(i_n)$. Therefore, we can bound Equation (E.55) simply by

$$\langle \Delta_{n,i_n}, \boldsymbol{v}_{k'} \rangle \leq \sqrt{2\varepsilon}. \tag{E.57}$$

We also need to find an upper bound for $\langle \Delta_{n,i_n}, \boldsymbol{w}_j^t \rangle$. Similar to previous analyses, we have

$$\begin{aligned}
\langle \Delta_{n,i_n}, \boldsymbol{w}_j^t \rangle &= \langle \boldsymbol{x}_{n,i_n}, \boldsymbol{w}_j^t \rangle - \sum_{j'\in\mathcal{S}_n(i_n)} \left\langle \sigma\left(\langle \boldsymbol{w}_{j'}^t, \boldsymbol{x}_{n,i_n}\rangle\right)\boldsymbol{w}_{j'}^t, \boldsymbol{w}_j^t \right\rangle \\
&\leq t\sigma_0 f_r + \sqrt{2\varepsilon}t\sigma_0 f_h + t\sigma_0 f_h \cdot \left(\sqrt{2\varepsilon}t\sigma_0 f_h + t\sigma_0 f_r\right)\cdot|\mathcal{S}_n^t(i_n)|.
\end{aligned} \tag{E.58}$$

Combine Equations (E.54) to (E.58), we finally get

$$\begin{aligned}
\frac{1}{2\eta_t}\langle \boldsymbol{w}_j^{t+1} - \boldsymbol{w}_j^t, \boldsymbol{v}_{k'} \rangle \leq{}& N\beta_{k'}(2\varepsilon + \sqrt{2\varepsilon})t\sigma_0 f_h \\
&+ \sqrt{2\varepsilon}N\beta_{k'}\left(t\sigma_0 f_r + \sqrt{2\varepsilon}t\sigma_0 f_h + t\sigma_0 f_h \cdot \left(\sqrt{2\varepsilon}t\sigma_0 f_h + t\sigma_0 f_r\right)\cdot|\mathcal{S}_n^t(i_n)|\right),
\end{aligned} \tag{E.59}$$

which can be further simplified to

$$\langle \boldsymbol{w}_j^{t+1}, \boldsymbol{v}_{k'} \rangle \leq 0.4N\beta_{k'}\sigma_0 \left( f_h \cdot (6\varepsilon + \sqrt{2\varepsilon} + \sqrt{2\varepsilon}t\sigma_0 f_h(c_2-1)d) + f_r \cdot \left( t\sigma_0 f_h \cdot (c_2-1)d + \sqrt{2\varepsilon} \right) \right).$$
(E.60)

As discussed in Appendix D, we can obtain

$$\langle \boldsymbol{w}_j^{t+1}, \boldsymbol{v}_{k'} \rangle \leq (t+1)\sigma_0 f_r$$
(E.61)

by setting parameters. Equations (E.48) and (E.60) together prove that $\boldsymbol{w}_j \in \tilde{\mathcal{S}}^{t+1}(k)$.

$\square$

*Remark* E.1 (Over-parameterized model is necessary). The "perfect reconstruction" given by Equation (3) could easily converge to local minima. According to the proof of Theorem 1, the member of each $\tilde{\mathcal{S}}(K)$ is decided by random initialization. It is possible that multiple kernels are included in one $\tilde{\mathcal{S}}(k)$. If $C = d$, then the kernels might not be able to learn all of the features. Therefore, to ensure (at least with high probability) that every feature with $\beta_k \neq 0$ is extracted by at least one of the kernels, we let $C > d$ in this paper. For simplicity, we let $C = c_2 d$ for some constant $c_2 > 0$.

*Corollary* 3.1. Consider two features $v_{k_1}$ and $v_{k_2}$ with $\beta_{k_1} > \beta_{k_2}$. For all $\alpha \in \{1, 2\}$, let $\boldsymbol{w}_{j_\alpha}$ be a kernel in the cone of $\boldsymbol{v}_{k_\alpha}$. Then, the bounds (cf. Equations (E.48) and (E.60)) for $\langle w_{j_1}, v_{k_1} \rangle - \langle w_{j_1}, v_{k'} \rangle$ (for all $k' \neq k_1$) is larger than those for $\langle w_{j_2}, v_{k_2} \rangle - \langle w_{j_2}, v_{k'} \rangle$ (for all $k' \neq k_2$).

*Theorem* 3. For $\theta \in (0, 1)$, $\forall j \in [C]$, and $\forall k \in [d]$, if $\langle \boldsymbol{w}_j, \boldsymbol{v}_k \rangle > (1 - \theta) \cdot \|\boldsymbol{w}_j\|$, then

$$\langle \boldsymbol{w}_j, \boldsymbol{x}_a \rangle < \|\boldsymbol{w}_j\| \cdot \max\{2\theta, \frac{1}{\theta\sqrt{d-1}}\}$$
(E.62)

hold with probability at least $2\theta \cdot \exp(-1/(2\theta^2))$

Theorem 3 can be viewed as the direct corollary of the geometrical inequalities in high-dimensional spaces (e.g., Lemma F.4). The proof of Theorem 3 is omitted here.

# F  TECHNICAL LEMMAS

The following two lemmas bound the upper and tail probability of i.i.d. random variables.

**Lemma F.1** (Upper tail). *Let $X_1, X_2, \cdots, X_L$ be i.i.d. random variables upper bounded by $M$. Denote $X = \sum_{i \in C}$. Then we have*

$$\boldsymbol{P}\left[X \geq \boldsymbol{E}\left[X\right] + \lambda\right] \leq \exp\left(-\frac{3\lambda^2}{6\sum_i \boldsymbol{E}(X_i^2) + 2M\lambda}\right)$$
(F.1)

*for all $\lambda > 0$.*

**Lemma F.2** (Lower tail). *Let $X_1, X_2, \cdots, X_L$ be i.i.d. random variables such that $X_i \leq -M$ for all $i \in [C]$. Denote $X = \sum_{i \in C}$. Then we have*

$$\boldsymbol{P}\left[X \leq \boldsymbol{E}\left[X\right] - \lambda\right] \leq \exp\left(-\frac{3\lambda^2}{6\sum_i \boldsymbol{E}(X_i^2) + 2M\lambda}\right)$$
(F.2)

*for all $\lambda > 0$.*

The proofs of Lemmas F.1 and F.2 can be found in various textbooks, which is omitted here. The following corollary is useful in Appendix E.

**Corollary F.1.** *Let $X_1, X_2, \cdots, X_C$ be i.i.d. Bernoulli variables such that $\boldsymbol{P}[X_i = 1] = p$ and $\boldsymbol{P}[X_i = 0] = 1 - p$ for all $i \in [C]$. Let $0 < p_1 < p_2 < 1$ be constants. If $p \in (p_1, p_2)$, then we have*

$$\boldsymbol{P}\left[\sum_{i \in [C]} X_i \leq Cp_1 - \lambda\right] \leq \exp\left(-\frac{\lambda^2}{2Cp_2}\right)$$
(F.3)

*and*

$$\boldsymbol{P}\left[\sum_{i \in [C]} X_i \geq Lp_2 + \lambda\right] \leq \exp\left(-\frac{3\lambda^2}{6Cp_2 + 2\lambda}\right)$$
(F.4)

*for all $\lambda > 0$.*

**Lemma F.3** (Concentration inequality for chi-squared random variables)**.**

$$\boldsymbol{P}\left[\left|\chi_d^2 - d\right| \geq \lambda d\right] \leq 2e^{-\lambda^2 d/8} \tag{F.5}$$

The proof of this lemma can be found in Example 2.11 of Wainwright (2019), which is omitted here.

The following lemma characterizes the concentration of probability on $d$-dimensional sphere.

**Lemma F.4.** *Let $X := (X_1, X_2, \cdots, X_d)$ be a random variable drawn from the uniform distribution on $\{\boldsymbol{x} : \|\boldsymbol{x}\| = 1\} \subset \mathbb{R}^d$. For all $c > 0$, the probability*

$$\boldsymbol{P}\left[X_1 > \frac{c}{\sqrt{d-1}}\right] \leq \frac{2}{c}e^{-\frac{c^2}{2}}. \tag{F.6}$$

The proof of this lemma can be found on the course website of Guruswami (2012).

## G EXPERIMENTS DETAILS

This section provides more details on the experiments and some auxiliary results.

### G.1 IMPLEMENTATION DETAILS

Both of the experiments are conducted on an Ubuntu 64-bit657 Linux workstation, having a 10-core Intel Xeon Silver CPU (2.20 GHz) and 4 Nvidia GeForce RTX658 2080 Ti GPUs with 11GB graphics memory.

### G.1.1 SYNTHESIZED EXPERIMENTS

All networks are initialized with Gaussian initialization and being trained for 1000 epochs to ensure convergence. We use stochastic gradient descent (SGD) with momentum = 0.9 and weight decay = 0.001. The model is defined in Section 2. The following algorithm specify the generating process of the normal data in the synthesized experiments.

---

**Algorithm 1** Generate normal data

---

**Require:** Sample size $N \in \mathbb{Z}^+$, features $\mathcal{V} = \mathcal{V}_{nor} \cup \mathcal{V}_{aux}$, total patches $P$, noise size $\varepsilon$
 1: **for** $n \in [N]$ **do**
 2:    $\mathcal{V}_{n,aux} \leftarrow$ randomly sample $P - 2$ features from $\mathcal{V}_{aux}$ without replacement
 3:    $\mathcal{V}_n \leftarrow \mathcal{V}_{n,aux} \cup \mathcal{V}_{nor}$
 4:    **for** $i \in [P]$ **do**
 5:       $\boldsymbol{v}_{k_i} \leftarrow$ randomly sampled from $\mathcal{V}_n$
 6:       $\rho \leftarrow$ randomly sampled from $\mathcal{N}(0, \varepsilon)$
 7:       $\boldsymbol{z}_n^{(i)} \leftarrow \boldsymbol{v}_{k_i} + \rho$
 8:    **end for**
 9:    **for** $\boldsymbol{v}_k \in \mathcal{V}$ **do**
10:       $\beta_k \leftarrow$ (total occurrence of $\boldsymbol{v}_k$ in $[\boldsymbol{z}_N]$) / $N$
11:    **end for**
12: **end for**
13: **return** $[\boldsymbol{z}_N], \beta$

---

### G.1.2 VISUALIZATION OF THE KERNELS

All networks are initialized with Gaussian initialization and trained for 100 epochs to ensure convergence. We use stochastic gradient descent (SGD) with momentum = 0.9 and weight decay = 0.001. The batch size is 32, and the learning rate is 0.001. To obtain images-like kernels, we visualize 24 kernels from the first layer.

**Dataset and network structure**    To be consistent with the theoretical analysis, we use Autoencoders to reconstruct the images from MNIST and CIFAR-10. We visualize the kernels in the first layer of the autoencoder.

## G.2 AUXILIARY EXPERIMENTS

This section presents some auxiliary experiments on the synthesized data to further validate our theoretical analysis in Section 3.

### G.2.1 HYPER-PARAMETERS STUDY

We mainly discuss the effect of over-parameterization and the number of replaced patches. In Section 2.2, we require the network to be over-parameterized. In Figure 4, we let $C = \text{int}(1.2d)$ and obtain desirable results. Here, we experiment on the under-parameterized setting with $C = \text{int}(0.8P)$ in comparison. The results are presented in Figure G.1. It can be observed that all the reconstruction errors increase significantly. Note that the scales of the $y$-axis are 0 to 0.02 in Figure G.1 and 0 to 0.014 in Figure 4. The comparative experiment implies that slight over-parameterization could help the model better reconstruct all types of inputs.

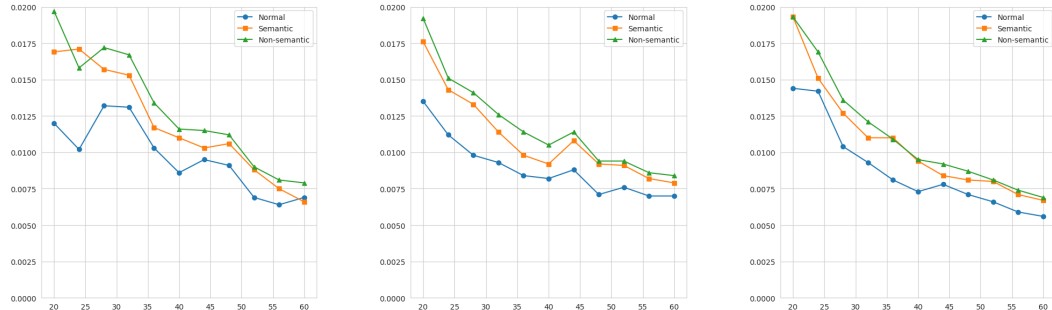

Figure G.1: Testing the under-parameterized setting with $C = \text{int}(0.8P)$. By comparing to Figure 4, this set of experimental results implies that slight over-parameterization (i.e., $C = \text{int}(1.2P)$)could help the model better reconstruct all types of inputs. Here, we primarily compare the magnitudes of $P$ and $C$. Specifically, the scenario where the ratio $C/P > 1$ is referred to as *over-parameterized*, while the converse (i.e., $C/P < 1$) is defined as *under-parameterized*.

In Section 3.2, we study the case when only one patch is replaced by the anomaly patch and claim that our analysis can be naturally extended to the case when multiple patches are replaced. We consider the *number of replaced patch* $\textbf{NRP} = \text{int}(c_4 P + 2)$ with $c_4 \in \{0, 0.1, 0.2\}$. As demonstrated in G.2, the gaps between the reconstruction errors of the normal data and the anomalies become significantly wider when more patches are replaced, which is consistent with our analysis.

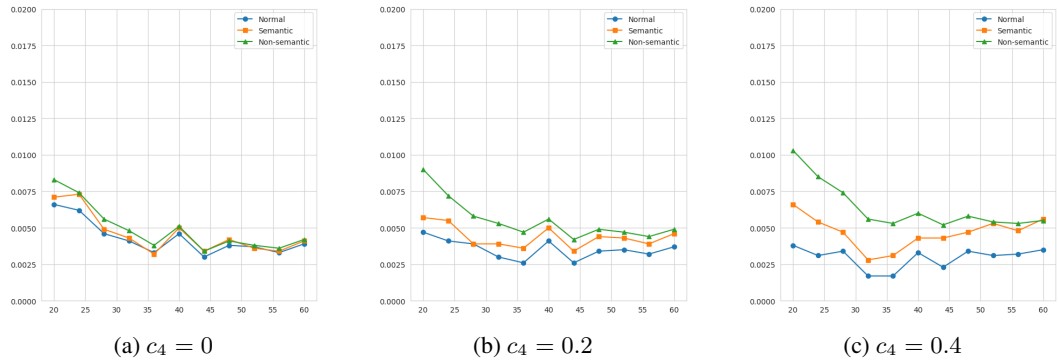

(a) $c_4 = 0$      (b) $c_4 = 0.2$      (c) $c_4 = 0.4$

Figure G.2: Testing different numbers of replaced patches. The parameter $c_4$ is used to characterize the number of patches replaced in anomalies, where a larger $c_4$ indicates a greater number of replaced patches. The gaps between the reconstruction errors of the normal data and the anomalies become significantly wider when more patches are replaced, which is consistent with our analysis.

### G.2.2 THE GROWTH OF THE NORM OF THE KERNELS

This section reports the growth of the norm of the kernels during training. According to Lemma 3.1, the growth rates of the norm of the kernels is upper bounded by Equations (11) and (12). As a side product of the synthesized experiments in Appendix G, we record the norm of the kernels during training. The results are presented in Figures G.3 - G.8. In all the experiments, the norm of the kernels increases almost linearly, which is consistent with our analysis in Equations (11) and (12). Additionally, kernels with larger initial values grow faster. It can be observed that as the number of kernels and patches increases (i.e., when the task becomes more complex), the differences between kernels also become more pronounced. The discussion of the results can be found in Remark G.1

### G.2.3 INNER PRODUCT BETWEEN THE KERNELS AND THE FEATURES

A key observation underlying our proposed cone set intuition is that kernels tend to converge toward features in terms of direction, which can be measured by the (normalized) inner product between the kernels and the features. We report the inner products between the kernels and the features during training in c The discussion of the results can be found in Remark G.1

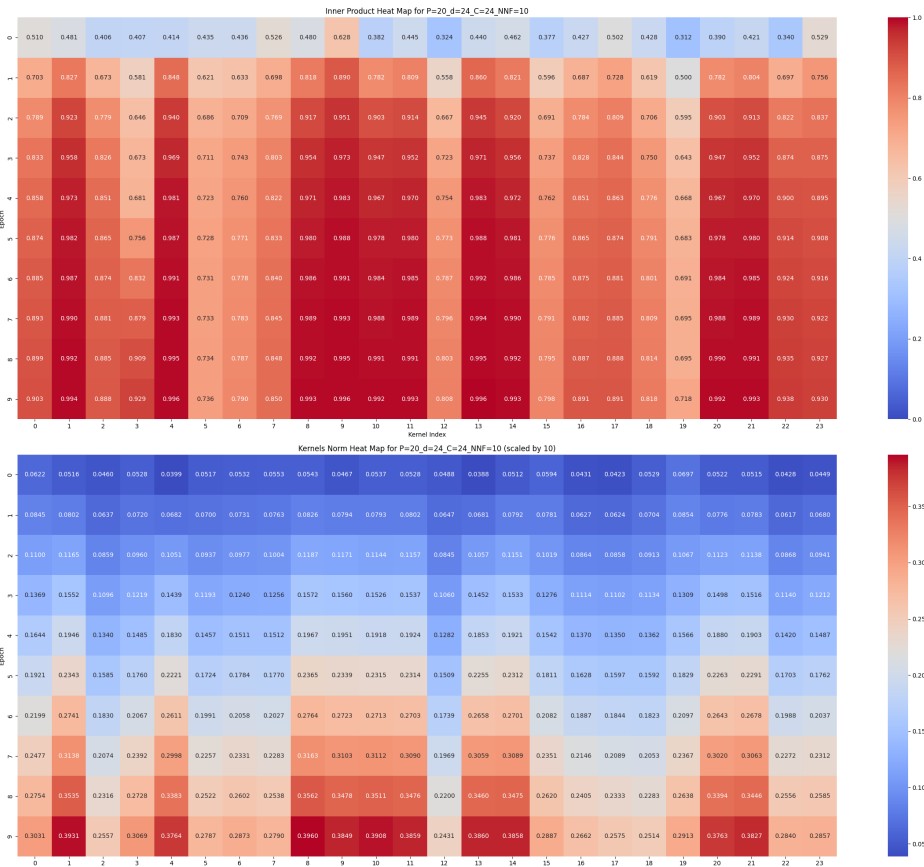

Figure G.3: Recorded **kernels' norm** (the *below* sub-figure) and **inner products between the kernels and the features** (the *above* sub-figure) during training. The controling variables are $P = 20, d = 24$, and $C = 24$. In both sub-figures, the $x$-axis represents the kernel index $j$ ($j \in [C]$), and the $y$-axis denotes the training epoch. For better visualization, we have scaled the norm of the kernels by a factor of 10 in the below sub-figure.

*Remark* G.1 (Discussions on the results in Appendices G.2.2 and G.2.3). An intuitive observation is that the color of each column (representing one single kernel) becomes progressively warmer from top to bottom in both sub-figures, which implies that both the norm and the inner product increase with the training epoch. In particular, the growth of the inner product can serve as an evidence

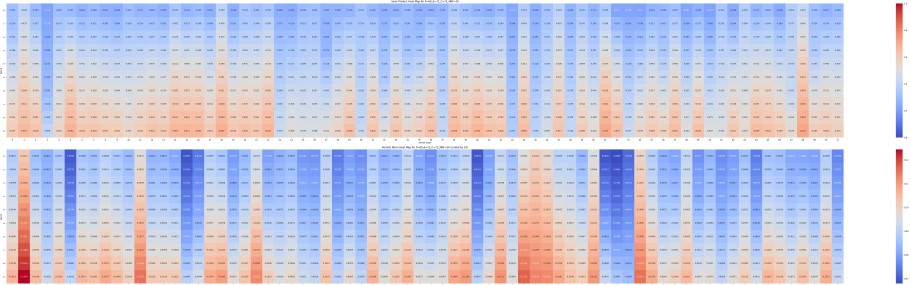

Figure G.4: The control variables are $P = 60, d = 72,$ and $C = 72$.

for our proposed cone set intuition, i.e., the kernels tend to converge toward features in terms of direction.

The main difference between Figures G.3 - G.8 lies in the controling variables $P, d,$ and $C$, which depict different levels of task complexity. As the dimension increases, randomly initialized kernels become more difficult to be absorbed (reflected in the bound provided by Theorem 2). Figures G.5 - G.8 provide more experimental results. The similar captions are omitted for brevity.

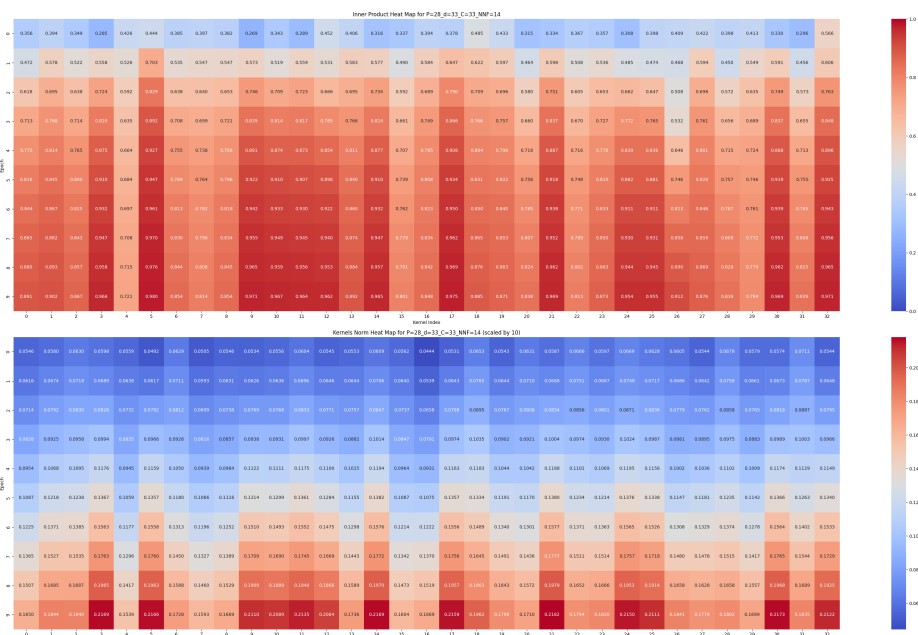

Figure G.5: The control variables are $P = 28, d = 33,$ and $C = 33$.

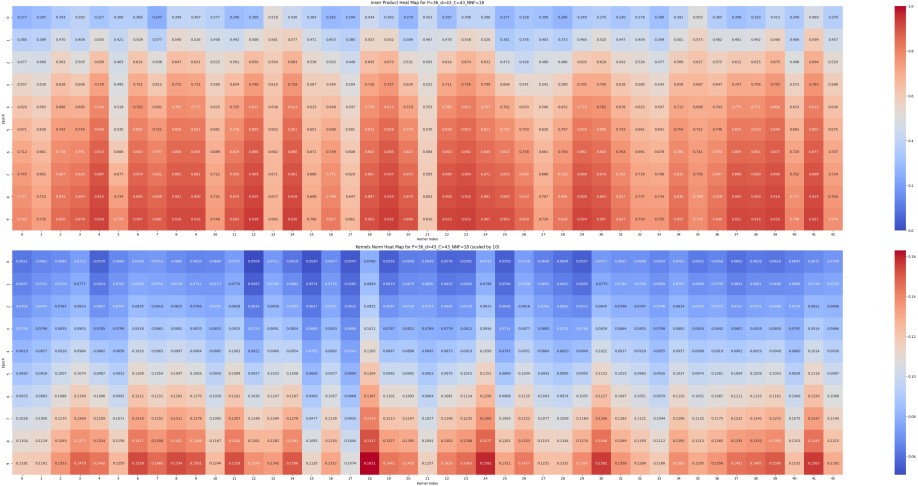

Figure G.6: The control variables are $P = 36, d = 43$, and $C = 43$.

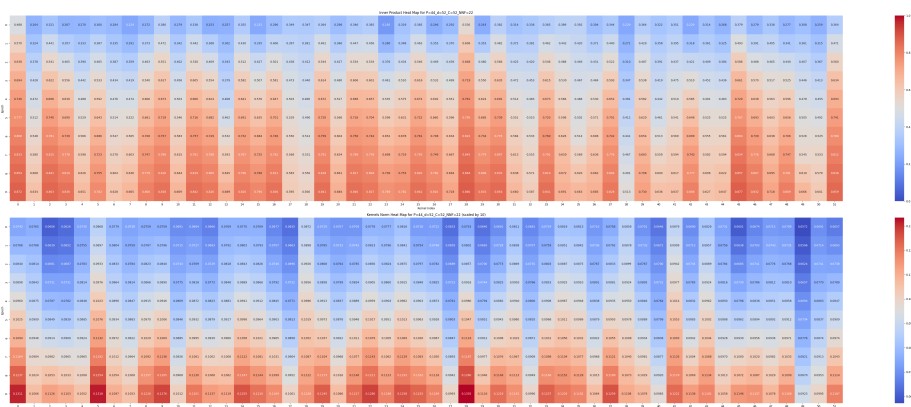

Figure G.7: The control variables are $P = 44, d = 52$, and $C = 52$.

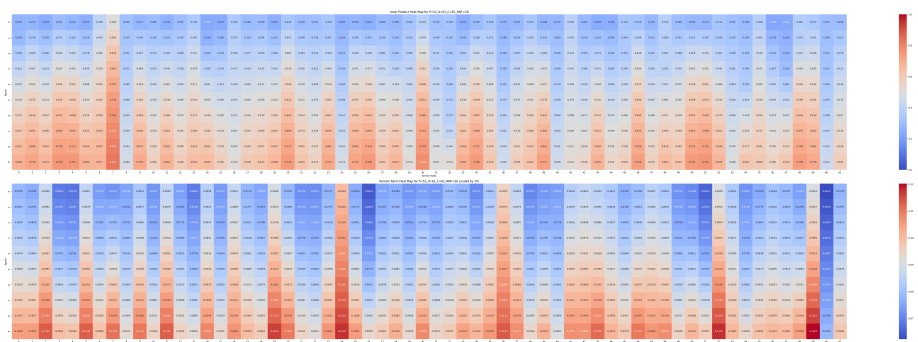

Figure G.8: The control variables are $P = 52, d = 62$, and $C = 62$.

### G.3 RAW RESULTS

This section reports the raw results of the synthesized experiments. The control variables (i.e., $\varepsilon$, $d$, and $C$) are noted in the captions of the tables.

Table 1: $\varepsilon = 0.01, d = \mathrm{int}(1.1P), C = \mathrm{int}(0.8P)$

| $(d, P, C)$ | NE $\pm$ Std. | SE $\pm$ Std. | NSE $\pm$ Std. |
|---|---|---|---|
| $(22, 20, 16)$ | $0.0120 \pm 0.0009$ | $0.0169 \pm 0.0011$ | $0.0197 \pm 0.0014$ |
| $(26, 24, 19)$ | $0.0102 \pm 0.0008$ | $0.0171 \pm 0.0012$ | $0.0158 \pm 0.0011$ |
| $(30, 28, 22)$ | $0.0132 \pm 0.0007$ | $0.0157 \pm 0.0009$ | $0.0172 \pm 0.0010$ |
| $(35, 32, 25)$ | $0.0131 \pm 0.0006$ | $0.0153 \pm 0.0007$ | $0.0167 \pm 0.0008$ |
| $(39, 36, 28)$ | $0.0103 \pm 0.0005$ | $0.0117 \pm 0.0005$ | $0.0134 \pm 0.0006$ |
| $(44, 40, 32)$ | $0.0086 \pm 0.0003$ | $0.0110 \pm 0.0005$ | $0.0116 \pm 0.0005$ |
| $(48, 44, 35)$ | $0.0095 \pm 0.0003$ | $0.0103 \pm 0.0004$ | $0.0115 \pm 0.0004$ |
| $(52, 48, 38)$ | $0.0091 \pm 0.0003$ | $0.0106 \pm 0.0004$ | $0.0112 \pm 0.0004$ |
| $(57, 52, 41)$ | $0.0069 \pm 0.0002$ | $0.0088 \pm 0.0003$ | $0.0090 \pm 0.0003$ |
| $(61, 56, 44)$ | $0.0064 \pm 0.0001$ | $0.0075 \pm 0.0002$ | $0.0081 \pm 0.0002$ |
| $(66, 60, 48)$ | $0.0069 \pm 0.0001$ | $0.0066 \pm 0.0001$ | $0.0079 \pm 0.0002$ |

Table 2: $\varepsilon = 0.01, d = \mathrm{int}(1.2P), C = \mathrm{int}(0.8P)$

| $(d, P, C)$ | NE $\pm$ Std. | SE $\pm$ Std. | NSE $\pm$ Std. |
|---|---|---|---|
| $(24, 20, 16)$ | $0.0135 \pm 0.0011$ | $0.0176 \pm 0.0014$ | $0.0192 \pm 0.0015$ |
| $(28, 24, 19)$ | $0.0112 \pm 0.0008$ | $0.0143 \pm 0.0011$ | $0.0151 \pm 0.0011$ |
| $(33, 28, 22)$ | $0.0098 \pm 0.0007$ | $0.0133 \pm 0.0009$ | $0.0141 \pm 0.0009$ |
| $(38, 32, 25)$ | $0.0093 \pm 0.0004$ | $0.0114 \pm 0.0005$ | $0.0126 \pm 0.0006$ |
| $(43, 36, 28)$ | $0.0084 \pm 0.0004$ | $0.0098 \pm 0.0004$ | $0.0114 \pm 0.0005$ |
| $(48, 40, 32)$ | $0.0082 \pm 0.0003$ | $0.0092 \pm 0.0003$ | $0.0105 \pm 0.0004$ |
| $(52, 44, 35)$ | $0.0088 \pm 0.0003$ | $0.0108 \pm 0.0004$ | $0.0114 \pm 0.0004$ |
| $(57, 48, 38)$ | $0.0071 \pm 0.0002$ | $0.0092 \pm 0.0003$ | $0.0094 \pm 0.0003$ |
| $(62, 52, 41)$ | $0.0076 \pm 0.0002$ | $0.0091 \pm 0.0002$ | $0.0094 \pm 0.0002$ |
| $(67, 56, 44)$ | $0.0070 \pm 0.0002$ | $0.0082 \pm 0.0002$ | $0.0086 \pm 0.0002$ |
| $(72, 60, 48)$ | $0.0070 \pm 0.0002$ | $0.0079 \pm 0.0002$ | $0.0084 \pm 0.0002$ |

Table 3: $\varepsilon = 0.01, d = \mathrm{int}(1.5P), C = \mathrm{int}(0.8P)$

| $(d, P, C)$ | NE $\pm$ Std. | SE $\pm$ Std. | NSE $\pm$ Std. |
|---|---|---|---|
| $(30, 20, 16)$ | $0.0144 \pm 0.0009$ | $0.0193 \pm 0.0013$ | $0.0193 \pm 0.0012$ |
| $(36, 24, 19)$ | $0.0142 \pm 0.0007$ | $0.0151 \pm 0.0008$ | $0.0169 \pm 0.0009$ |
| $(42, 28, 22)$ | $0.0104 \pm 0.0004$ | $0.0127 \pm 0.0005$ | $0.0136 \pm 0.0005$ |
| $(48, 32, 25)$ | $0.0093 \pm 0.0003$ | $0.0110 \pm 0.0004$ | $0.0121 \pm 0.0004$ |
| $(54, 36, 28)$ | $0.0081 \pm 0.0003$ | $0.0110 \pm 0.0004$ | $0.0109 \pm 0.0004$ |
| $(60, 40, 32)$ | $0.0073 \pm 0.0002$ | $0.0094 \pm 0.0003$ | $0.0095 \pm 0.0003$ |
| $(66, 44, 35)$ | $0.0078 \pm 0.0002$ | $0.0084 \pm 0.0002$ | $0.0092 \pm 0.0002$ |
| $(72, 48, 38)$ | $0.0071 \pm 0.0002$ | $0.0081 \pm 0.0002$ | $0.0087 \pm 0.0002$ |
| $(78, 52, 41)$ | $0.0066 \pm 0.0002$ | $0.0080 \pm 0.0002$ | $0.0081 \pm 0.0002$ |
| $(84, 56, 44)$ | $0.0059 \pm 0.0001$ | $0.0071 \pm 0.0002$ | $0.0074 \pm 0.0002$ |
| $(90, 60, 48)$ | $0.0056 \pm 0.0001$ | $0.0067 \pm 0.0001$ | $0.0069 \pm 0.0001$ |

Table 4: $\varepsilon = 0.01, d = \text{int}(1.2P), C = \text{int}(1.2d)$

| $(d, P, C)$ | NE ± Std. | SE ± Std. | NSE ± Std. |
|---|---|---|---|
| $(24, 20, 28)$ | $0.0084 \pm 0.0006$ | $0.0128 \pm 0.0011$ | $0.0140 \pm 0.0011$ |
| $(28, 24, 33)$ | $0.0073 \pm 0.0004$ | $0.0111 \pm 0.0006$ | $0.0124 \pm 0.0008$ |
| $(33, 28, 39)$ | $0.0044 \pm 0.0003$ | $0.0049 \pm 0.0003$ | $0.0089 \pm 0.0005$ |
| $(38, 32, 45)$ | $0.0040 \pm 0.0002$ | $0.0058 \pm 0.0003$ | $0.0077 \pm 0.0004$ |
| $(43, 36, 51)$ | $0.0043 \pm 0.0002$ | $0.0057 \pm 0.0002$ | $0.0077 \pm 0.0003$ |
| $(48, 40, 57)$ | $0.0042 \pm 0.0001$ | $0.0051 \pm 0.0002$ | $0.0072 \pm 0.0003$ |
| $(52, 44, 62)$ | $0.0050 \pm 0.0002$ | $0.0053 \pm 0.0002$ | $0.0074 \pm 0.0003$ |
| $(57, 48, 68)$ | $0.0048 \pm 0.0002$ | $0.0058 \pm 0.0002$ | $0.0070 \pm 0.0002$ |
| $(62, 52, 74)$ | $0.0049 \pm 0.0001$ | $0.0059 \pm 0.0002$ | $0.0066 \pm 0.0002$ |
| $(67, 56, 80)$ | $0.0044 \pm 0.0001$ | $0.0051 \pm 0.0001$ | $0.0061 \pm 0.0002$ |
| $(72, 60, 86)$ | $0.0036 \pm 0.0001$ | $0.0044 \pm 0.0001$ | $0.0053 \pm 0.0001$ |

Table 5: $\varepsilon = 0.01, d = \text{int}(1.5P), C = \text{int}(1.2d)$

| $(d, P, C)$ | NE ± Std. | SE ± Std. | NSE ± Std. |
|---|---|---|---|
| $(30, 20, 36)$ | $0.0030 \pm 0.0002$ | $0.0046 \pm 0.0004$ | $0.0086 \pm 0.0006$ |
| $(36, 24, 43)$ | $0.0074 \pm 0.0004$ | $0.0083 \pm 0.0005$ | $0.0107 \pm 0.0006$ |
| $(42, 28, 50)$ | $0.0039 \pm 0.0001$ | $0.0041 \pm 0.0001$ | $0.0069 \pm 0.0003$ |
| $(48, 32, 57)$ | $0.0035 \pm 0.0001$ | $0.0049 \pm 0.0002$ | $0.0067 \pm 0.0003$ |
| $(54, 36, 64)$ | $0.0025 \pm 0.0001$ | $0.0032 \pm 0.0001$ | $0.0052 \pm 0.0002$ |
| $(60, 40, 72)$ | $0.0029 \pm 0.0001$ | $0.0036 \pm 0.0001$ | $0.0052 \pm 0.0001$ |
| $(66, 44, 79)$ | $0.0034 \pm 0.0001$ | $0.0040 \pm 0.0001$ | $0.0053 \pm 0.0001$ |
| $(72, 48, 86)$ | $0.0030 \pm 0.0001$ | $0.0039 \pm 0.0001$ | $0.0049 \pm 0.0001$ |
| $(78, 52, 93)$ | $0.0039 \pm 0.0001$ | $0.0047 \pm 0.0001$ | $0.0053 \pm 0.0001$ |
| $(84, 56, 100)$ | $0.0041 \pm 0.0001$ | $0.0053 \pm 0.0001$ | $0.0057 \pm 0.0001$ |
| $(90, 60, 108)$ | $0.0031 \pm 0.0001$ | $0.0039 \pm 0.0001$ | $0.0044 \pm 0.0001$ |

Table 6: $\varepsilon = 0.1, d = \text{int}(2P), C = \text{int}(1.2d)$

| $(d, P, C)$ | NE ± Std. | SE ± Std. | NSE ± Std. |
|---|---|---|---|
| $(40, 20, 48)$ | $0.0049 \pm 0.0002$ | $0.0060 \pm 0.0002$ | $0.0091 \pm 0.0004$ |
| $(48, 24, 57)$ | $0.0046 \pm 0.0001$ | $0.0061 \pm 0.0002$ | $0.0077 \pm 0.0003$ |
| $(56, 28, 67)$ | $0.0042 \pm 0.0001$ | $0.0045 \pm 0.0001$ | $0.0062 \pm 0.0002$ |
| $(64, 32, 76)$ | $0.0033 \pm 0.0001$ | $0.0046 \pm 0.0001$ | $0.0056 \pm 0.0002$ |
| $(72, 36, 86)$ | $0.0027 \pm 0.0001$ | $0.0037 \pm 0.0001$ | $0.0048 \pm 0.0001$ |
| $(80, 40, 96)$ | $0.0042 \pm 0.0001$ | $0.0051 \pm 0.0001$ | $0.0057 \pm 0.0001$ |
| $(88, 44, 105)$ | $0.0024 \pm 0.0000$ | $0.0033 \pm 0.0000$ | $0.0040 \pm 0.0001$ |
| $(96, 48, 115)$ | $0.0034 \pm 0.0001$ | $0.0045 \pm 0.0001$ | $0.0048 \pm 0.0001$ |
| $(104, 52, 124)$ | $0.0036 \pm 0.0000$ | $0.0044 \pm 0.0001$ | $0.0048 \pm 0.0001$ |
| $(112, 56, 134)$ | $0.0033 \pm 0.0000$ | $0.0041 \pm 0.0000$ | $0.0045 \pm 0.0001$ |
| $(120, 60, 144)$ | $0.0038 \pm 0.0000$ | $0.0047 \pm 0.0001$ | $0.0049 \pm 0.0001$ |

Table 7: $\varepsilon = 0.01, d = \text{int}(2P), C = \text{int}(1.2d)$

| $(d, P, C)$ | NE $\pm$ Std. | SE $\pm$ Std. | NSE $\pm$ Std. |
|---|---|---|---|
| $(40, 20, 48)$ | $0.0047 \pm 0.0002$ | $0.0057 \pm 0.0002$ | $0.0090 \pm 0.0004$ |
| $(48, 24, 57)$ | $0.0041 \pm 0.0001$ | $0.0055 \pm 0.0002$ | $0.0072 \pm 0.0003$ |
| $(56, 28, 67)$ | $0.0039 \pm 0.0001$ | $0.0039 \pm 0.0001$ | $0.0058 \pm 0.0002$ |
| $(64, 32, 76)$ | $0.0030 \pm 0.0001$ | $0.0039 \pm 0.0001$ | $0.0053 \pm 0.0001$ |
| $(72, 36, 86)$ | $0.0026 \pm 0.0001$ | $0.0036 \pm 0.0001$ | $0.0047 \pm 0.0001$ |
| $(80, 40, 96)$ | $0.0041 \pm 0.0001$ | $0.0050 \pm 0.0001$ | $0.0056 \pm 0.0001$ |
| $(88, 44, 105)$ | $0.0026 \pm 0.0000$ | $0.0034 \pm 0.0000$ | $0.0042 \pm 0.0001$ |
| $(96, 48, 115)$ | $0.0034 \pm 0.0001$ | $0.0044 \pm 0.0001$ | $0.0049 \pm 0.0001$ |
| $(104, 52, 124)$ | $0.0035 \pm 0.0000$ | $0.0043 \pm 0.0001$ | $0.0047 \pm 0.0001$ |
| $(112, 56, 134)$ | $0.0032 \pm 0.0000$ | $0.0039 \pm 0.0000$ | $0.0044 \pm 0.0001$ |
| $(120, 60, 144)$ | $0.0037 \pm 0.0000$ | $0.0046 \pm 0.0001$ | $0.0049 \pm 0.0001$ |

Table 8: $\varepsilon = 0.001, d = \text{int}(2P), C = \text{int}(1.2d)$

| $(d, P, C)$ | NE $\pm$ Std. | SE $\pm$ Std. | NSE $\pm$ Std. |
|---|---|---|---|
| $(40, 20, 48)$ | $0.0047 \pm 0.0002$ | $0.0057 \pm 0.0002$ | $0.0090 \pm 0.0004$ |
| $(48, 24, 57)$ | $0.0041 \pm 0.0001$ | $0.0055 \pm 0.0002$ | $0.0072 \pm 0.0003$ |
| $(56, 28, 67)$ | $0.0039 \pm 0.0001$ | $0.0039 \pm 0.0001$ | $0.0058 \pm 0.0002$ |
| $(64, 32, 76)$ | $0.0030 \pm 0.0001$ | $0.0039 \pm 0.0001$ | $0.0053 \pm 0.0001$ |
| $(72, 36, 86)$ | $0.0026 \pm 0.0001$ | $0.0036 \pm 0.0001$ | $0.0047 \pm 0.0001$ |
| $(80, 40, 96)$ | $0.0041 \pm 0.0001$ | $0.0050 \pm 0.0001$ | $0.0056 \pm 0.0001$ |
| $(88, 44, 105)$ | $0.0026 \pm 0.0000$ | $0.0034 \pm 0.0000$ | $0.0042 \pm 0.0001$ |
| $(96, 48, 115)$ | $0.0034 \pm 0.0001$ | $0.0044 \pm 0.0001$ | $0.0049 \pm 0.0001$ |
| $(104, 52, 124)$ | $0.0035 \pm 0.0000$ | $0.0043 \pm 0.0001$ | $0.0047 \pm 0.0001$ |
| $(112, 56, 134)$ | $0.0032 \pm 0.0000$ | $0.0039 \pm 0.0000$ | $0.0044 \pm 0.0001$ |
| $(120, 60, 144)$ | $0.0037 \pm 0.0000$ | $0.0046 \pm 0.0001$ | $0.0049 \pm 0.0001$ |

