# OpenReview forum: "Two-Layer Convolutional Autoencoders Trained on Normal Data Provably Detect Unseen Anomalies"
_ICLR.cc/2026/Conference — ICLR 2026 Poster_

### Official Review · Reviewer_2q5h · 2025-10-26

**Soundness:** 2
**Presentation:** 2
**Contribution:** 4
**Rating:** 6
**Confidence:** 4

**Summary:**

The paper addresses the insufficient theoretical understanding of reconstruction-based anomaly detection (RBAD) and proposes a new theoretical framework introducing the concept of a “cone set” to describe the dynamics of feature learning during the training of convolutional autoencoders (CAEs). Through rigorous derivations, the authors demonstrate how convolution kernel parameters are gradually attracted to the “cone set” and align with corresponding true feature directions. Overall, the theoretical contribution lies in establishing a formal analytical foundation for RBAD, filling a major gap in the literature where theoretical explanations for autoencoder-based anomaly detection were largely absent.

**Strengths:**

1. The paper’s greatest strength lies in constructing a systematic theoretical framework for RBAD. While previous works achieved strong empirical performance, they lacked a clear explanation for why reconstruction error distinguishes anomalies. Through the cone set and feature absorption mechanism, this paper is the first to explain—via gradient dynamics—why AEs tend to retain only normal features while poorly reconstructing anomalies. The discussion in VPDM [1] of the “identical shortcut” phenomenon indirectly supports this view: if a model relies excessively on raw input rather than stable feature extraction, it may reconstruct anomalies as well, leading to detection failure. The cone set theory precisely explains that models which genuinely learn features, rather than memorizing inputs, naturally exhibit poor reconstruction for anomalies.
2. The theoretical formulation is highly structured, with well-defined assumptions, lemmas, and proofs. The use of Hilbert space representations for patches, smooth ReLU activations for differentiability, and global max pooling to avoid gradient entanglement all demonstrate careful mathematical design.
3. The paper’s theoretical results align with long-standing community observations. For instance, it formally explains why RBAD struggles with semantic anomalies (since they may contain normal local features) and why mild overparameterization helps avoid suboptimal reconstructions. These insights transform empirical heuristics into theoretically grounded knowledge, strengthening both credibility and interpretability.

[1] Li Y, Feng Y, Chen B, et al. Vague Prototype-Oriented Diffusion Model for Multi-Class Anomaly Detection

**Weaknesses:**

1. The theoretical analysis relies on several idealized assumptions. For example, the data are assumed to consist of P patches, each containing one dominant feature plus random noise. Real images are more complex, patches may not be independent and could contain multiple mixed features. Similarly, the definition of anomalies as “patch replacements” may not cover global distribution shifts or complex anomaly patterns.
2. The paper only analyzes a two-layer convolutional autoencoder (single hidden convolutional layer + pooling). In practice, anomaly detection models are often deeper and structurally richer. The conclusions may not directly generalize to these more complex architectures, and care must be taken when extrapolating.
3. The main limitation lies in the narrow scope of empirical validation. Although the paper includes some experiments, they are limited compared to the complexity of anomaly detection tasks. The synthetic experiments, while rigorous, remain toy settings far from real-world conditions.
4. The paper barely discusses or compares with current state-of-the-art (SOTA) anomaly detection methods. While pure theory need not compete in performance, the omission creates a sense of detachment: the theory explains a relatively basic method, while the field’s focus has partially shifted toward more advanced ones. This gap could reduce confidence in the theory’s practical relevance.

**Questions:**

1. Could the theoretical assumptions be relaxed, or could the authors discuss whether similar principles hold for other generative models (e.g., diffusion models, GANs)? Additionally, if multiple patches are replaced (non-local anomalies), would the conclusions still hold?
2. Given the theoretical weakness of RBAD on semantic anomalies, could auxiliary tasks or regularization be introduced to enforce lower reconstruction ability for non-primary features?
3. The experimental section should be expanded.
4. In the revised version, clarify the relationship between this paper’s theoretical focus and existing empirical advances—emphasize that the goal is to complement, not compete with, SOTA detection methods.

---

> ### Author Response · Authors · 2025-11-28
> **Rebuttal to Reviewer 2q5h (1/2)**
>
> We would like to express our sincere gratitude to the reviewer for their valuable comments and constructive feedback. We have prepared detailed responses to each of the reviewer’s concerns below.
>
> > **Since we have revised the manuscript based on the reviewer’s guidance, all line numbers referenced herein correspond to those in the revised version. The important changes are marked in red.**
>
> **Response to Weakness 1.** It is unavoidable for a theoretical work to adopt specific assumptions on network structure and data model. The following table lists some of the previous works on feature learning and their assumptions.
>
> |Publication | Network Structure | Data Model|
> |:----|:--------|:-------|
> |Ours | 2-layer CNN autoencoder | patch-like data|
> |Allen-Zhu and Li (2021) | 2-layer fully connected network | sparse coding model (SCM)|
> |Allen-Zhu and Li (2023) | 2-layer CNN | patch-like data|
> |Jelassi et al (2023) | ViT with two simplified attention layers | images with sparse spatial connectivity patterns|
> |Chen et al (2023) | 2-layer CNN | modified SCM|
> |Collins et al (2024) | 2-layer fully connected network | modified SCM|
>
> Table R.1. Previous works on feature learning and their assumptions
>
> In particular, we elaborate the two assumptions highlighted by the reviewers as follows.
>
> 1. *(The P-Patch assumption)* The P-patch assumption is widely adopted in the feature learning community. While real-world data is indeed “more complex and patches may not be independent and could contain multiple mixed features”, we have elaborated on the rationality of our "overlapping patch + max pooling" assumption in Appendix B (Lines 629-651). In brief, we argue that the max-pooling operation serves to select “the most representative patch among the nearby overlapping patches”. Our simplifying assumption essentially embeds the selection process into the assumption itself.
>
> 2. *(The patch replacement assumption)* This assumption is motivated by some widely adopted AD datasets in industrial flaw detection (e.g., MVTec AD) and decease detection (e.g., Retinal-OCT).  In these scenarios, anomalies typically manifest as flaws (e.g., burnt spots or cracks) or lesions that can be captured in one single patch, which aligns with our "patch replacement" hypothesis for the anomalies. Admittedly, our assumption cannot encompass "global distribution shifts or complex anomaly patterns," but it retains adequate representativeness.
>
> **Response to Weakness 2.** As the name suggests, feature learning theory primarily focuses on the dynamics of *feature extraction* in neural networks. While deeper and structurally richer networks are indeed more commonly used, we argue that some add-on structures (e.g., layer-norm and batch-norm) are employed to enhance the training efficiency and robustness by alleviating the issues such as gradient vanishing, gradient exploding, and network degradation. These structures do not fundamentally change the feature learning process. As for the depth, it is a common practice in learning theory community to analyze two-layer networks for some explanatory results. Since both multi-layer and two-layer networks update their parameters via back-propagation, analyses conducted on two-layer networks also exhibits certain representativeness.
>
> We fully understand and concur with the reviewer’s point that "care must be taken when extrapolating." However, existing learning theory cannot analyze problems in an "end-to-end" manner as is commonly practiced in real-world applications. Consistent with most contemporary works in this field, we provide theoretical analysis of a specific sub-module under reasonable assumptions, and we believe this contributes to the advancement of the field.

---

> ### Author Response · Authors · 2025-11-28
> **Rebuttal to Reviewer 2q5h (2/2)**
>
> **Response to Weaknesses 3 & 4**. Since the main contribution of our paper lies in theoretical analyses, the experiments we conducted aim at complementing our theory and illustrating our intuition, which cannot match the complexity of SOTA RBAD experiments. To avoid misinterpretation, we have strengthened in the revised paper that “the goal of our paper is to complement, not compete with, SOTA detection methods” in Lines 99-100 as requested by the reviewer in Question 4. The following discussions are included in the revised paper.
>
> - In the learning theory community, it is a common practice to conduct synthetic experiments under the same setting as the theoretical analysis, serving to demonstrate whether the theoretical modeling can effectively characterize the practical problems (i.e., RBAD). In the caption of Figure 4, we briefly explain how the experimental results are consistent with our theory (cf. Response to Weakness 2) and the existing results (cf. Lines 38-43). More specifically, Lines 446-447 explain that the synthetic experiments conducted under our theoretical setting exhibit clear reconstruction error gaps between normal data and the anomalies, which is a key rationale underlying RBAD.
>
> - We use experiments on real-world datasets to illustrate the proposed cone set intuition. As a supplement to the existing explanations (e.g., Figures 2, 3, and Remark 3.3), the experiments in Figure 5 discuss the manifestations of the cone set intuition (cf. Lines 452-255) on networks trained with MNIST and CIFAR-10, which indirectly verifies the validity of our theory.
>
> **Response to Question 1.** Certain aspects of our theory can be relaxed. As noted in Lines 378–384, our theoretical framework can be trivially extended to scenarios involving the replacement of multiple patches, given that the mean squared error (Equation (7)) we consider is patch-wise additive. On the other hand, during the brief rebuttal period, it is hard to determine whether a similar cone set intuition is applicable to diffusion models and GANs, considering that the proving techniques for diffusion models and GANs are significantly different to ours.
>
> **Response to Question 2.** Intuitively, it seems feasible to reduce the model's learning efficiency for non-primary features (i.e., auxiliary features) by designing or introducing specific tasks or regularization. This, in turn, will decrease the reconstruction efficiency of semantic anomalies, thereby enhancing the success rate of anomaly detection. In practical applications, however, learning certain auxiliary features is sometimes unavoidable. For instance, the fur of some cat breeds may resemble that of other animals, as exemplified by an anomaly detection task where the normal data consist of cat images. Therefore, we suggest that *appropriate data cleaning and prior predictions of potential anomalies are likely to be effective approaches*. Nevertheless, this lies beyond the scope of the present study.
>
> **Response to Questions 3&4.** We are deeply grateful to the reviewer for their valuable recommendations and perceptive comments, which have helped us further refine our work. We have revised the paper based on the reviewer’s guidance and marked the important changes by red.
>
> ---
>
> Thank you again for your diligent efforts in reviewing our paper. Please do not hesitate to inform us if we have adequately addressed your concerns. Any further feedback is greatly appreciated, and we look forward to your reply.
>
> **References**
>
> [1] *Feature Purification: How Adversarial Training Performs Robust Deep Learning*, **Allen-Zhu and Li**, FOCS 2021.
>
> [2] *Benign Overfitting for Two-layer ReLU Convolutional Neural Networks*, **Kou et al**, ICML 2023
>
> [3] *Provable Guarantees for Neural Networks via Gradient Feature Learning*, **Shi et al**, NeurIPS 2023
>
> [4] *Contamination-resilient anomaly detection via adversarial learning on partially-observed normal and anomalous data*, **Lv et al**, ICML 2024
>
> [5] *Vague prototype-oriented diffusion model for multi-class anomaly detection*, **Li et al**, ICML 2024
>
> [6] *Mvtec ad–a comprehensive real-world dataset for unsupervised anomaly detection*, **Bergmann et al.**, CVPR 2019.

---

### Official Review · Reviewer_Vbxc · 2025-10-29

**Soundness:** 3
**Presentation:** 2
**Contribution:** 4
**Rating:** 6
**Confidence:** 3

**Summary:**

This paper proposes a new anomaly detection method, which provides a theoretical analysis of reconstruction-based
anomaly detection (RBAD). It constructs a two-layer convolutional autoencoder to reconstruct data and proves reconstructing normal data is easier than anomalies.

**Strengths:**

- This paper proposes an autoencoder model to reconstruct the normal data and anomalies, which trains on normal data. And it shows that the reconstruction error of normal data is smaller than that of anomalies.
- Theoretical analysis support the observation and conclusion of this paper.
- Experimental results on synthesis dataset validate the theoretical findings.

**Weaknesses:**

- How to use the non-semantic anomaly and semantic anomaly during training phase.
-The organization of section 3 is a bit chaotic. It is unclear to me how this method works.
- I am confused with the experimental results on real data. I don't know how to observe Figure 5. Please provide more explanations.
- Why not detect anomalies directly based on the real datasets. Then we can see quantitative results of Acc and F1.

**Questions:**

See Weaknesses.

---

> ### Author Response · Authors · 2025-11-28
> **Rebuttal to Reviewer Vbxc**
>
> We would like to express our sincere gratitude to the reviewer for their valuable comments and constructive feedback. We have prepared detailed responses to each of the reviewer’s concerns below.
>
> > **Since we have revised the manuscript based on the reviewer’s guidance, all line numbers referenced herein correspond to those in the revised version. The important changes are marked in red.**
>
> **Response to Weakness 1.** As stated in Line 176, both semantic and non-semantic anomalies are not considered in the training phase of RBAD. This constitutes one of the primary advantages of RBAD, as users typically lack prior knowledge of anomaly-related information in the training phase. We have emphasized this in the revised paper in Lines 175-177.
>
> In Lines 37-41, we briefly explain the workflow of RBAD. The target network in RBAD is a generative model trained exclusively on normal data to minimize the reconstruction error. An *intriguing observation* is that generative models trained on normal data tend to produce larger reconstruction errors when reconstructing anomalies than normal data. Anomalies can be identified due to their greater reconstruction errors.
>
> The primary goal of this paper is to theoretically explain this intriguing observation. Section 3 is devoted to theoretical analysis and presents the corresponding results. We analyze the training and reconstruction phases separately in Sections 3.1 and 3.2. The main conclusions and corresponding explanations are outlined in Figure 2.
>
> **Response to Weakness 2.** The experimental results in Figure 5 are presented to illustrate our proposed cone set intuition. Take the results on MNIST (Figure 5.(a)-5.(c)) as an example. The visualized kernels are randomly initialized at epoch 0. It can be observed in Figure 5.(a) that the kernels are completely noisy. By comparing Figures 5.(b) and 5.(c), the following two observations are closely related to our proposed theory.
>
> 1. Some kernels (e.g., (row-2,col-3) and (row-3, col-4)) exhibit clear contours at the early stage of training (epoch 10) and do not undergo significant changes at convergence (epoch 50).
> 2. Meanwhile, some other kernels (e.g., (row-1, col-1) and (row-3, col-2)) fail to exhibit distinct outlines even after convergence, which aligns with the case when the kernel is not absorbed by any of the cones.
> Both of the above points are consistent with the probability-based training dynamics proposed in our work. As a supplement to the existing explanations (e.g., Figures 2, 3, and Remark 3.3), these experiments discuss the manifestations of the cone set intuition (cf. Lines 452-255) on networks trained with MNIST and CIFAR-10, which indirectly verifies the validity of our theory. We have integrated the above discussions into the revised paper (Lines 1208-1217).
>
> **Response to Weakness 3.** Since the main contribution of our paper lies in theoretical analysis, developing a novel anomaly detection algorithm falls outside of the scope of our paper. On one hand, empirical results of RBAD have been reported by many existing works ([1], [2]), and we don’t have to re-validate these results. On the other hand, empirical works of RBAD generally lack rigorous proofs. Given the inherently black-box nature of neural networks and other AI technologies, interpretability-oriented theoretical research is valuable for deepening our understanding of the technology and safeguarding its robustness. In summary, we respectfully argue that our theoretical contributions do not require additional empirical validation on real-world datasets.
>
> ---
>
> Thank you again for your diligent efforts in reviewing our paper. Please do not hesitate to inform us if we have adequately addressed your concerns. Any further feedback is greatly appreciated, and we look forward to your reply.
>
> **References**
>
> [1] *Contamination-resilient anomaly detection via adversarial learning on partially-observed normal and anomalous data*, **Lv et al**, ICML 2024
>
> [2] *Vague prototype-oriented diffusion model for multi-class anomaly detection*, **Li et al**, ICML 2024

---

### Official Review · Reviewer_Po5J · 2025-10-30

**Soundness:** 3
**Presentation:** 3
**Contribution:** 3
**Rating:** 6
**Confidence:** 3

**Summary:**

This paper gives a theory for why reconstruction-based anomaly detection (RBAD) succeeds on normal data yet struggles on anomalies by analyzing a two-layer convolutional autoencoder with max pooling.
It introduces cone sets to show that training aligns kernels to frequent normal features.
This alignment yields weak activations and large reconstruction errors for non-semantic anomalies,
while yields small reconstruction errors for semantic anomalies containing learned features.
The proposed theory is validated on both synthetic and real datasets.

**Strengths:**

- Although reconstruction-error–based anomaly detection is widely used, there has been little theoretical analysis of why normal data are well reconstructed while anomaly data are not. This paper provides a theoretical explanation for this.
- The paper is well written and uses figures effectively to explain complex theory, making it very easy to follow.

**Weaknesses:**

Please refer to the Questions section for details.

**Questions:**

- While this theory analyzes autoencoders, I am curious what role it would play for variational autoencoders (VAEs).
As generative models, VAEs assign high likelihood to normal data and low likelihood to anomalies.
That is, they reconstruct normal data well and fail to reconstruct anomalies.
However, as noted in [1],
VAEs can sometimes assign higher likelihood to anomaly data than to normal data.
Since a VAE can be viewed as a regularized autoencoder,
I wonder whether this theory applies.
Could this theory help explain that phenomenon?
- Could this theory also be effective for more complex architectures, such as ResNet-based autoencoders?

[1] Nalisnick, Eric, et al. "Do deep generative models know what they don't know?." arXiv preprint arXiv:1810.09136 (2018).

---

> ### Author Response · Authors · 2025-11-28
> **Rebuttal to Reviewer Po5J**
>
> We would like to express our sincere gratitude to the reviewer for their valuable comments and constructive feedback. We have prepared detailed responses to each of the reviewer’s concerns below.
>
> > **Since we have revised the manuscript based on the reviewer’s guidance, all line numbers referenced herein correspond to those in the revised version. The important changes are marked in red.**
>
>
> **Response to Questions 1 & 2.** We sincerely thank the reviewer for pointing out potential future directions and providing relevant references. In the feature learning community, a single paper typically focuses on a specific structure or problem. The following table lists some of the previous works on feature learning and their adopted models.
>
>
> |Publication | Network Structure | Data Model|
> |:----|:--------|:-------|
> |Ours | 2-layer CNN autoencoder | patch-like data|
> |Allen-Zhu and Li (2021) | 2-layer fully connected network | sparse coding model (SCM)|
> |Allen-Zhu and Li (2023) | 2-layer CNN | patch-like data|
> |Jelassi et al (2023) | ViT with two simplified attention layers | images with sparse spatial connectivity patterns|
> |Chen et al (2023) | 2-layer CNN | modified SCM|
> |Collins et al (2024) | 2-layer fully connected network | modified SCM|
> Table R.1. Previous works on feature learning and their assumptions
>
> Given the rigorousness required for theoretical proofs, it is typically impractical to incorporate various model structures, e.g., VAEs and ResNet-based AEs. More specifically,
>
> 1. Compared to our considered model, VAE introduces probabilistic latent spaces and typically incorporates an ELBO term into their loss functions, which invalidates our definitions of encoder (Equation (1)), decoder (Equation (6)), and loss function (Equation (7)), together with the subsequent theoretical deduction. As stated in Lines 296-297, after random initialization, the weights of our network iterate in a deterministic manner, as is the case with most networks (e.g., ResNet and Transformers, under the condition that dropout is not employed). Therefore, analyses towards VAEs generally require different proof techniques than ours. Our theory cannot be trivially extended to explain the phenomenon mentioned by the reviewer, while the derivation related to VAEs falls beyond the scope of our work.
>
> 2. As noted earlier, a typical research paper in feature learning focuses on a single structural design; consequently, ResNet-based AEs also fall beyond the scope of our discussion. Besides, a defining design of ResNet-based networks is the residual blocks integrated with skip connections, whose primary function is to alleviate gradient vanishing and gradient exploding. These problems are essential in the training of deep networks; nevertheless, to the best of our knowledge, the structure of skip connections does not fundamentally change the underlying feature learning process of the network.
>
> Thank you again for your diligent efforts in reviewing our paper. Please do not hesitate to inform us if we have adequately addressed your concerns. Any further feedback is greatly appreciated, and we look forward to your reply.

---

### Official Review · Reviewer_HVHx · 2025-10-31

**Soundness:** 3
**Presentation:** 3
**Contribution:** 2
**Rating:** 4
**Confidence:** 3

**Summary:**

This paper investigates the behavior of the convolution kernels in reconstruction-based anomaly detection (RBAD). To derive theoretical results, this paper assumes that image data can be decomposed into normal/auxiliary patch features and the model is a two-layer CNN. Under these assumptions, this paper reveals that the kernels are absorbed into the cone set oriented toward the normal feature. The paper claims that it can explain the reconstruction error of anomaly data becomes large and verifies theoretical results through synthetic/real-world data.

**Strengths:**

1. This paper is well written except for the section of experiment.
2. This seems the first theoretical explanation for reconstruction-based anomaly detection.
3. The absorption into the cone-set appears to be an interesting and reasonable.
4. Several theoretical results might inspire researchers to propose a new anomaly detection method or to improve existing one.

**Weaknesses:**

1. Experiments need more explanation and discussions. Section 4 does have the explanation about Figure 4.

2. Empirical evaluations seem insufficient. Which theoretical results do Fig. 4 and Fig. 5 verify?
I suspect there some theoretical results might be verifiable. For example, since Lemma 3.1 indicates that the norm of kernel increases according to the training step,
it can be verifiable by plotting the norm vs training steps. I think the cone set is also verifiable by investigating values of the inner product in Def.3.1.
If not, why is it difficult and how do Fig. 4 and Fig. 5 support theoretical results?

3. Since fully-connected layers are also commonly used for anomaly detection in AE, a comparison with them might be interesting.
For a fully-connected layer, if the input vector contains a normal feature linearly independent of the noise component, intuitively one would expect the weight matrix to have singular vectors in the direction of the normal feature. This would result in singular values of zero for the anomaly feature, leading to large reconstruction errors.
Considering that convolution is a special case of a linear layer [a,b], a similar argument holds. Under such assumptions, it seems likely that the frequency components of an image would be formulated as the normal feature. The kernel visualization in Fig. 5 might also support such explanation as each kernel possesses a specific spectrum.
Does the proposed theory have any advantages or more reasonable insights over such approach?

[a] Tsuzuku, Y., and Sato, I.  "On the structural sensitivity of deep convolutional networks to the directions of fourier basis functions". CVPR2019

[b] Sedghi, H., and et al. "The singular values of convolutional layers". ICLR2018

4. The contribution can be a bit weak if theoretical findings can only explalin the reactions to semantic anomalies or the magnitude of reconstruction errors for anomalies.
This paper would be strengthened if results can suggest directions for reconstruction-based anomaly detection researches or methods for improvement.
Can theoretical results suggest such directions or methods?

**Questions:**

Could you read Weakness and answer the questions?

---

> ### Author Response · Authors · 2025-11-28
> **Rebuttal to Reviewer HVHx (1/2)**
>
> We would like to express our sincere gratitude to the reviewer for their valuable comments and constructive feedback. We have prepared detailed responses to each of the reviewer’s concerns below.
>
> > **Since we have revised the manuscript based on the reviewer’s guidance, all line numbers referenced herein correspond to those in the revised version. The important changes in the revised paper are marked in red.**
>
> **Response to Weakness 1 (Explanation and discussions on the experimental results).** Since the main contribution of our paper lies in theoretical analyses, the explanation and discussions on experimental results are intentionally structured to focus on the following two aspects:
>
> - In the learning theory community, it is a common practice to conduct synthetic experiments under the same setting as the theoretical analysis, serving to demonstrate *whether the theoretical modeling can effectively characterize the practical problems (i.e., RBAD)*. In the caption of Figure 4, we briefly explain how the experimental results are consistent with our theory (cf. Response to Weakness 2) and the existing results (cf. Lines 38-43). More specifically, Lines 446-447 explain that the synthetic experiments conducted under our theoretical setting exhibit clear reconstruction error gaps between normal data and the anomalies, which is a key rationale underlying RBAD.
>
> - We use experiments on real-world datasets to *illustrate the proposed cone set intuition*. As a supplement to the existing explanations (e.g., Figures 2, 3, and Remark 3.3), the experiments in Figure 5 discuss the manifestations of the cone set intuition (cf. Lines 452-255) on networks trained with MNIST and CIFAR-10, which indirectly verifies the validity of our theory.
>
> Due to ICLR’s strict page limit (even for the revised PDF), in this stage, we can only provide the most concise explanations and discussions on the experimental results in the main text. We have included the above discussions in Appendix G (Lines 1192-1217) in the revised paper and will incorporate additional discussions mentioned in this comment into the main text in future revisions.
>
> **Response to Weakness 2. (On our empirical evaluation)** We have emphasized in the revised paper (Lines 99-100) that *our goal is to complement, not compete with, SOTA RBAD methods* (this comment is raised by Reviewer 2q5h, which aptly summarizes the nature of our work). In Line 422, we state that the experiments validate our theoretical findings, which refer to the results we summarized in the theoretical roadmap (i.e., Figure 2). As illustrated in Figure 2, Theorems 1-3 collectively characterize the complex feature learning dynamics of RBAD, which we term the cone set intuition. Individually, none of these three theorems is particularly significant; they only become intuitive when considered together. Returning to the reviewer’s question, we clarify our empirical evaluation as follows.
>
> - *(Which theoretical results do Figures 4 and 5 verify?)* We have explained the focus of the experiments in our reply to Weakness 1: Figure 4 serves to demonstrate whether our theoretical modeling can effectively characterize RBAD, and Figure 5 is used to illustrate the cone set intuition on real-world datasets. It is difficult to specify which exact theorem or lemma is directly verified by Figures 4 and 5 since, as we noted earlier, none of these theorems is particularly significant if considered separately. A reasonable response to this question is that Figures 4 and 5 verify the cone set intuition.
>
> - *(Which of the theoretical results are verifiable?)* As speculated, many of the theoretical results are indeed verifiable, including Lemma 3.1. However, we argue that verifying Theorems 1-3 in isolation (e.g., focusing solely on the growth of the norm of the weights) offers limited value for elaborating on the cone set intuition; in fact, it may even misdirect readers’ attention. Nevertheless, we have still conducted auxiliary experiments regarding the growth of the norm of the kernels during training. The results are presented in Section G.3 (Lines 1276-1451).
>
> - *(Verifying the cone set intuition by investigating the value of the inner product)* We sincerely thank the reviewer for their constructive suggestions. During the rebuttal period, we conducted auxiliary experiments using the same setting as the synthesized experiments in Section 4 and recorded the inner products of the kernels and the features. The results are discussed in Section G.3 in the revised submission. Under various settings, the inner product exhibits a distinct decreasing trend during training, which further validates our cone set intuition.
>
> In summary, our approach is to validate the cone set intuition as a whole, rather than verifying individual theorems or lemmas in isolation. We have conducted auxiliary experiments to further validate the cone set intuition in Appendix G.3 in the revised paper.

---

> ### Author Response · Authors · 2025-11-28
> **Rebuttal to Reviewer HVHx (2/2)**
>
> **Response to Weakness 3. (Comparison to fully-connected layers)** We sincerely thank the reviewer for pointing out potential future directions and providing relevant references. Within the feature learning community, many existing works have explored the training dynamics of fully-connected networks ([1]-[3]). These studies generally employ proof techniques that are similar to ours yet distinct in key aspects. Given the rigorousness required for theoretical proofs, it is typically impractical for theoretical works to incorporate various model structures. As such, *comparisons with fully-connected networks fall outside the scope of our study*.
>
> We have carefully read references (1) and (2) in the review. However, the techniques employed in these works are quite distinct from those adopted in our study, given the different research focuses and problem settings. During the brief rebuttal period, it is hard to determine whether our theory offers any advantages or more reasonable insights over the approaches related to fully-connected networks.
>
> **Response to Weakness 4. (Suggestions for future directions of RBAD)** We sincerely thank the reviewer for their constructive suggestions. The core contribution of this paper lies in its theoretical analyses. In contrast, certain pioneering works on RBAD ([4], [5]) lack rigorous proofs. Given the inherently black-box nature of neural networks and other AI technologies, interpretability-oriented theoretical research is of great value for deepening our understanding of the technology and safeguarding its robustness. Our work is motivated by the insights from empirical studies and underpinned by rigorous theoretical proofs, thereby contributing to the advancement of this field.
>
> ---
>
> Thank you again for your diligent efforts in reviewing our paper. Please do not hesitate to inform us if we have adequately addressed your concerns. Any further feedback is greatly appreciated, and we look forward to your reply.
>
> **References**
>
> [1] *Feature Purification: How Adversarial Training Performs Robust Deep Learning*, **Allen-Zhu and Li**, FOCS 2021.
>
> [2] *Benign Overfitting for Two-layer ReLU Convolutional Neural Networks*, **Kou et al**, ICML 2023
>
> [3] *Provable Guarantees for Neural Networks via Gradient Feature Learning*, **Shi et al**, NeurIPS 2023
>
> [4] *Contamination-resilient anomaly detection via adversarial learning on partially-observed normal and anomalous data*, **Lv et al**, ICML 2024
>
> [5] *Vague prototype-oriented diffusion model for multi-class anomaly detection*, **Li et al**, ICML 2024

---

### Meta-Review · Area_Chair_K9RX · 2026-01-06

**Summary:**

This paper provides a learning-theoretic analysis of reconstruction-based anomaly detection (RBAD) using a two-layer convolutional autoencoder. The core idea is the “cone set” intuition: during training on normal data, convolutional kernels align with frequent normal features, leading to small reconstruction error on normal data and larger errors on anomalies. Across reviewers, there is broad agreement that the theoretical contribution is novel and fills a gap in the literature. The main disagreements are about (i) how convincing and well-explained the experiments are, (ii) whether the scope (two-layer CNN, patch-based assumptions) is too narrow, and (iii) whether the paper should do more to connect to SOTA RBAD methods or practical anomaly detection benchmarks.

**Reviewer Concerns:**

1. Experimental clarity and interpretation (what exactly Figures 4 and 5 verify): partially addressed.
The rebuttal adds detailed explanations and appendix experiments (kernel norm growth, inner products) and clarifies that experiments are meant to validate the cone set intuition as a whole, not individual theorems. This improves clarity, but some reviewers may still find the validation indirect.

2. Limited empirical evaluation / lack of quantitative real-data benchmarks: partially addressed.
The authors clearly restate that the goal is theory, not competing with SOTA, and justify why synthetic and illustrative real-data experiments are sufficient. This addresses intent, but does not change the fact that empirical scope remains narrow.

3. Strong modeling assumptions (patch replacement, two-layer CNN, max pooling): partially addressed.
The rebuttal gives reasonable justification and context, pointing to standard practice in learning theory and added discussion in appendices. However, the assumptions are still idealized, and generalization to deeper or more complex models remains speculative.

4. Relationship to fully connected AEs, VAEs, deeper models, or modern RBAD methods: addressed at a conceptual level.
The rebuttal is clear that these are out of scope and would require different proof techniques. This is a reasonable boundary-setting, though it does not strengthen practical relevance.

5. Contribution strength / actionable insights for improving RBAD: Not fully addressed.
The rebuttal argues that interpretability and theoretical grounding are the contribution. It does not really translate theory into concrete algorithmic guidance, which some reviewers hoped for.

**Reviewer Scores:**

Reviewer HVHx (4 => 4/5): Experimental explanations and auxiliary experiments help, but core concerns about scope and contribution strength largely remain.

Reviewer Po5J (6 => 6): Questions were mostly about extensions (VAEs, ResNets); rebuttal clarifies scope but doesn’t change fundamentals.

Reviewer Vbxc (6=>6): Rebuttal improves clarity around Figure 5 and training setup; still no new real-data quantitative results.

Reviewer 2q5h (6=>6): Rebuttal directly addresses most weaknesses with clearer positioning and added discussion, but empirical scope remains limited.

---

### Decision · Program_Chairs · 2026-01-26

Accept (Poster)